# TwinVoice: A Multi-dimensional Benchmark Towards Digital Twins via LLM Persona Simulation

## Abstract

Large Language Models (LLMs) are exhibiting emergent human-like abilities and are increasingly envisioned as the foundation for simulating a specific communication style, behavioral tendencies, and personality traits. However, current evaluations of LLM-based persona simulation remain limited: most rely on synthetic dialogues, lack systematic frameworks, and lack analysis of the capability requirement. To address these limitations, we introduce TwinVoice, a comprehensive benchmark for assessing persona simulation across diverse real-world contexts. TwinVoice encompasses three dimensions: Social Persona (public social interactions), Interpersonal Persona (private dialogues), and Narrative Persona (role-based expression). The ability of LLMs in persona simulation is further decomposed into six fundamental capabilities, including opinion consistency, memory recall, logical reasoning, lexical fidelity, persona tone, and syntactic style. Experimental results reveal that while advanced models achieve moderate accuracy, they remain insufficient in sustaining consistent persona simulation, especially lacking the capability of syntactic style and memory recall. Our data, code, and evaluation results are available at `https://anonymous.4open.science/r/TwinVoice-B08E`.

## 1 Introduction

Large Language Models (LLMs) are rapidly evolving from basic text generators into human-like agents (Bubeck et al., 2023; Wei et al., 2022; Chang et al., 2024). Existing studies have shown that the most advanced LLMs are capable of producing text indistinguishable from human writing (Jones & Bergen, 2025; Jones et al., 2025; Jones & Bergen, 2024). Consequently, the research focus is shifting toward a highly specific challenge: *Can we construct "digital twins" of specific individuals that are indistinguishable from themselves?* To address this challenge, the primary technical path is through LLM-based persona simulation, which replicates a person's unique style of talking, behavior, and personality (Shanahan et al., 2023; Park et al., 2023) based on their data. LLM-based persona simulation is supposed to unlock a series of applications, including highly personalized assistants (Ma et al., 2023; Li et al., 2025a), social simulations (Li et al., 2023; Ran et al., 2025), healthcare (Barricelli et al., 2020), and marketing (Hornik & Rachamim, 2025). Despite growing interest in creating digital twins with LLM-based persona simulation, its current ability remains unexplored due to the lack of systematic evaluation (Toubia et al., 2025; Zhou et al., 2025).

To address this issue, current evaluations have tried to test LLM's ability in imitating and predicting human behaviors. For example, BehaviorChain (Li et al., 2025b) evaluates continuous persona-based behavior by requiring models to iteratively predict the next action given persona profile and history, with performance degrading as chains lengthen. Human Simulacra and PersoBench assess human-likeness and personalized response quality, while other studies probe persona-driven decision making, counterfactual adherence, and large-scale dynamic profiling (Xie et al., 2025; Afzoon et al., 2024; Xu et al., 2024; Kumar et al., 2025; Jiang et al., 2025). However, those evaluation benchmarks face limitations in both their scope and granularity. On the one hand, the predominant reliance on synthetic dialogues (Shen et al., 2023; Tu et al., 2024) prevents benchmarks from capturing the rich expression of human identity across diverse real-world contexts (**Scope Limitation**). On the other hand, current benchmarks are often evaluated based on an LLM's accuracy in predicting

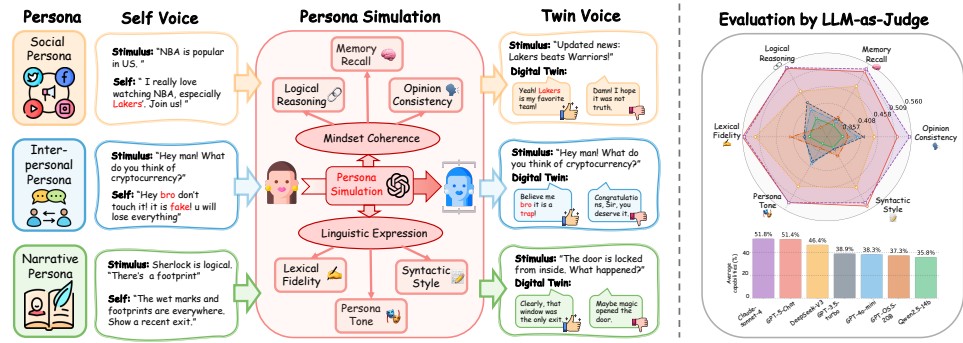

Figure 1: **The conceptual framework of TwinVoice: (Left)** The evaluation is structured across three core **dimensions** that represent distinct aspects of identity expression: *Social Persona* (public facing), *Interpersonal Persona* (private interaction), and *Narrative Persona* (fictional scenarios). The LLMs are prompted with a person's historical context to simulate their behavior. The LLM's ability for persona simulation is categorized into six fundamental **capabilities**. **(Right)** Experimental results averaged over three dimensions are presented.

human behavior, leaving a critical gap in understanding the fundamental capabilities—such as memory, reasoning, and lexical fidelity—that a model must possess for authentic simulation (*Granularity Limitation*).

To bridge the gap between the vision of digital twins and the current capabilities of persona simulation, we introduce **TwinVoice**, a comprehensive benchmark designed for realistic and fine-grained persona evaluation (see Table 1 for a comparison with prior persona-simulation benchmarks; "persona size" denotes the number of distinct, independent personas per benchmark). To address the scope limitation, TwinVoice is grounded in both real-world and synthetic data across three complementary dimensions in persona simulation (see Figure 1): **Social Persona**, **Interpersonal Persona**, **Narrative Persona**. The **Social Persona** dimension leverages real-world social media data to evaluate a public-facing identity, while the **Interpersonal Persona** dimension utilizes multi-session dialogue data to assess a more private, relational self. While these two dimensions are grounded in authentic digital footprints, the **Narrative Persona** is designed to complement such data with fictional scenarios to test behaviors and narrative consistency in more diverse contexts. Addressing the granularity limitations of holistic accuracy evaluations, we shift from end-to-end scoring to capability-level assessment. Building on psycholinguistic evidence that language conveys both what people say and how they say it (Pennebaker et al., 2003), we group persona fidelity into **Mindset Coherence** and **Linguistic Expression.** Mindset Coherence assesses the logical and factual consistency of the content, including Opinion Consistency (Zaller, 1992), Memory Recall (Clark & Brennan, 1991), and Logical Reasoning (Kahneman, 2011). Linguistic Expression evaluates the language's stylistic form, encompassing Lexical Fidelity (Mehl et al., 2006; Koppel et al., 2009), Persona Tone (Brown, 1987), and Syntactic Style (Biber, 1995). To obtain objective, low-variance accuracy with controlled distractors, we use a discriminative multiple-choice setting. To capture the open-ended persona consistency required by real digital twins, we adopt a generative setting and evaluate outputs with an LLM-as-a-Judge in ranking and scoring modes, with a human agreement check (see Sections 5.3 and 5.4).

Table 1: A comparison of TwinVoice with Prior LLM Persona-Simulation Benchmarks.

| Benchmark | Persona Size | Real-World Sourcing | Multiple Dimensions | Multi-Paradigm Evaluation | Human Baseline | Fine-Grained Capabilities | Multilingual Coverage |
|---|---|---|---|---|---|---|---|
| Human Simulacra (Xie et al., 2025) | 11 | ✗ | ✗ | ✗ | ✓ | ✗ | ✗ |
| BehaviorChain (Li et al., 2025b) | 1,001 | ✓ | ✗ | ✓ | ✗ | ✗ | ✗ |
| PersonaEval (Zhang et al.) | 130 | ✓ | ✓ | ✗ | ✓ | ✓ | ✗ |
| PERSONAMEM (Jiang et al., 2025) | 20 | ✗ | ✓ | ✓ | ✗ | ✓ | ✗ |
| **TwinVoice** [OURS] | **4,553** | ✓ | ✓ | ✓ | ✓ | ✓ | ✓ |

We test a series of state-of-the-art LLMs on TwinVoice and reveal several key insights into current capabilities and limitations in persona simulation with LLMs. On discriminative accuracy, GPT-3.5-Turbo averages 47.5%, while advanced models reach 71.2% for GPT-5 and 76.2% for Claude-

Sonnet-4 (Anthropic, 2025). In the generative setting with an LLM-as-a-Judge, GPT-5 (OpenAI, 2025) leads with 48.5% judged accuracy and a 2.13 pairwise score, with Claude-Sonnet-4 close at 47.9% and 2.12. To validate the Judge and clarify model versus human performance, we conduct two targeted human annotations: (i) a discriminative Dimension 1 subset of 50 items, and (ii) a generative evaluation for ranking and scoring. In the discriminative study, majority vote accuracy is 66.0%, GPT-5 reaches 60.0%, and model versus human agreement is high ($\kappa$=0.634). In the generative study, human versus Judge agreement is high as well ($\kappa$=0.646 for ranking; Spearman $\rho$=0.591 for scores). As for dimensions, performance is highest under the Narrative persona, while Social and Interpersonal lag. Across capabilities, models perform best on Lexical Fidelity and Opinion Consistency and worst on Persona Tone and Memory Recall. Performance dispersion across LLMs is large for all capabilities, indicating high discriminative power. These patterns will guide subsequent research and upgrades to LLM persona simulation.

Contributions of this work are threefold: **(1)** We introduce TwinVoice, a comprehensive benchmark for evaluating LLM-based persona simulation across multiple real-world scenarios with systematic competency decomposition; **(2)** We develop novel evaluation methodologies combining discriminative assessment with LLM-as-Judge for generative tasks; and **(3)** We provide extensive empirical analysis showing the limitations of the most advanced LLMs in person simulation and offer crucial insights for advancing personalized AI systems.

## 2    RELATED WORK

### 2.1    PERSONALIZED AGENTS AND DIGITAL TWINS

The construction of digital twins, virtual replicas of specific individuals, is an emerging challenge in AI (Shanahan et al., 2023; Park et al., 2023). Originating in engineering as counterparts to physical systems (Grieves & Vickers, 2017), the concept now extends to AI agents that capture a person's communication style, preferences, and personality. Recent efforts have operationalized this vision across diverse domains. Examples include reviving anime characters (Li et al., 2023), simulating agent societies from novels (Ran et al., 2025), and evaluating impersonation of writing styles and memories (Shi et al., 2025). Applications have been explored in healthcare (Barricelli et al., 2020), marketing (Hornik & Rachamim, 2025), and through industry systems like SecondMe (Shang et al., 2024) for lifelong personal modeling. While these human-centered digital twins promise highly personalized chatbots (Ma et al., 2023; Li et al., 2025a) and ubiquitous computing applications (Fast et al., 2016), prior research has often focused narrowly on style imitation, overlooking the broader competencies required for authentic persona simulation.

### 2.2    DATASETS, BENCHMARKS, AND EVALUATION FOR PERSONA SIMULATION

Progress in this area depends on high-quality datasets and benchmarks. Recent resources have begun to fill this gap, offering diverse evaluation protocols. Benchmarks have been developed from large-scale surveys of human traits (Toubia et al., 2025; Chen et al., 2025), persona-based behavior chains (Li et al., 2025b), psychology-guided agent evaluations (Xie et al., 2025), persona-driven decision-making tasks (Afzoon et al., 2024; Xu et al., 2024), and multi-party dialogue role identification (Zhou et al., 2025). More recent work explores challenging settings like counterfactual simulation (Kumar et al., 2025) and dynamic user profiling (Jiang et al., 2025).

Despite this growing landscape, evaluations remain fragmented and often rely on synthetic data, limiting their ecological validity. This highlights the need for a unified framework to advance digital twin research rigorously. Our TwinVoice benchmark addresses these limitations by leveraging real-world social media, conversational, and fictional data to provide authentic and systematic evaluation across multiple persona dimensions.

## 3    TASK FORMULATION

### 3.1    PROBLEM DEFINITION

TwinVoice evaluates LLMs' ability to simulate human personas through a unified task paradigm that captures the essence of digital twin functionality. Formally, we define the persona simulation task as follows:

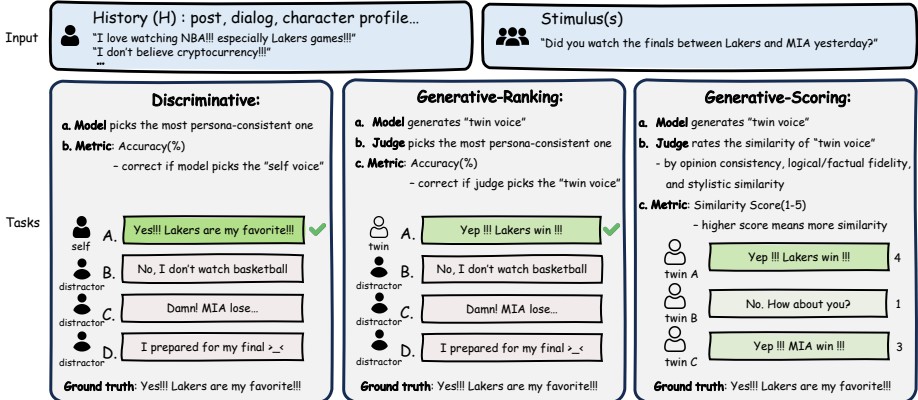

Figure 2: **TwinVoice experiment evaluation overview: Top**: The LLMs are prompted with a specific persona's history and tasked with a stimulus. **Bottom**: Three evaluation protocols: Discriminative: the model chooses among A–D, one of which is the ground truth persona behavior. Generative-Ranking: the model writes and an LLM-as-Judge selects the best candidate, yielding Acc.(Gen). Generative–Scoring: the model writes and the Judge rates similarity on opinion, logic, and style, yielding Score(Gen).

Given a persona's historical data $\mathcal{H} = \{h_1, h_2, \ldots, h_n\}$ and a current stimulus $s$, the history is instantiated per dimension (Social, Interpersonal, or Narrative) as social posts, multi-session conversations, or narrative materials, respectively. The objective is to generate a response $r$ that maximally approximates the ground truth response $r^*$ that the original persona would produce in stimulus $s$, which can be formulated as an optimization problem:

$$r^* = \arg\max_r P(r|\mathcal{H}, s, \theta_{\text{persona}}),\tag{1}$$

where $\theta_{\text{persona}}$ represents the latent persona characteristics learned from historical data $\mathcal{H}$. The evaluation objective is to assess how well an LLM $M$ can approximate this optimal response:

$$\text{Score} = f_{\text{sim}}(M(\mathcal{H}, s), r^*),\tag{2}$$

where $f_{\text{sim}}$ denotes a similarity function that measures persona consistency across multiple dimensions.

TwinVoice instantiates this general framework across three dimensions, each defined by its history source and interaction stimulus:

---

**Persona Dimensions**

**Social Persona.** In this dimension, $\mathcal{H}$ consists of a user's historical social media posts $\mathcal{H}^{\text{social}} = \{h_1^{(social)}, h_2^{(social)}, \ldots, h_m^{(social)}\}$, and the stimulus $s$ represents a new post requiring a response. The challenge lies in maintaining stylistic consistency and opinion alignment in public discourse.

**Interpersonal Persona.** Here, $\mathcal{H}$ comprises multi-session conversational history $\mathcal{H}^{\text{inter}} = \{h_1^{(inter)}, h_2^{(inter)}, \ldots, h_k^{(inter)}\}$ where each $h_i^{(inter)}$ represents a dialogue session. The stimulus $s$ is a new utterance from a conversation partner, requiring the model to generate contextually appropriate responses while maintaining conversational authenticity and memory-grounded consistency.

**Narrative Persona.** In this dimension, $\mathcal{H}$ encompasses character background information and behavioral records $\mathcal{H}^{\text{narra}} = \{h_1^{(narra)}, h_2^{(narra)}, \ldots, h_l^{(narra)}\}$ where each $h_i^{(narra)}$ denotes either background information or a prior action. The stimulus $s$ describes a narrative scenario requiring character reaction, testing the model's ability to maintain role-based expression fidelity.

---

Across all three settings, we adopt a capability-centric evaluation rather than a single holistic score. The decomposition and scoring criteria are detailed in Section 4.2.

## 3.2 EVALUATION METHODOLOGIES

To balance objectivity and ecological validity, we pair a discriminative multiple-choice evaluation (objective, low-variance accuracy under controlled distractors) with a generative evaluation (open-ended persona fidelity via LLM-as-a-Judge in ranking and scoring).

### 3.2.1 DISCRIMINATIVE EVALUATION

The discriminative evaluation transforms the generation task into a multiple-choice selection problem. For each test instance $(s, r^*)$, we construct a candidate set $\mathcal{C} = \{r^*, r_1, r_2, r_3\}$ where $r^*$ is the ground truth response and $\{r_1, r_2, r_3\}$ are distractors. The evaluated LLM must select the most persona-consistent response from the shuffled candidate set.

The construction of distractors varies across dimensions to ensure realistic evaluation scenarios:

---

**Distractor Construction**

**Social Persona:** Distractors are sampled from authentic responses by other users to similar posts, preserving topical relevance while introducing stylistic and opinion variations.

**Interpersonal Persona:** Distractors are selected from real conversational responses in similar contexts, maintaining conversational appropriateness while differing in personal characteristics.

**Narrative Persona:** Distractors are generated using advanced LLMs with alternative character interpretations, ensuring narrative coherence while diverging from the target persona's behavioral patterns.

---

Discriminative evaluation provides direct accuracy measurements:

$$\text{Accuracy} = \frac{1}{N} \sum_{i=1}^{N} \mathbf{1}[M(\mathcal{H}_i, s_i) = r_i^*], \tag{3}$$

where $N$ is the total number of test instances and $\mathbf{1}[\cdot]$ is the indicator function.

### 3.2.2 GENERATIVE EVALUATION

While discriminative evaluation offers clear interpretability, real-world digital twin applications require open-ended generation capabilities. Our generative evaluation employs LLM-as-a-Judge Gu et al. (2024); Ye et al. (2025) protocols to assess response quality along multiple dimensions.

We implement two distinct judging approaches:

**Scoring-based Evaluation.** The judge model rates generated responses against ground truth using structured evaluation criteria. Given a stimulus $s$, generated response $r_{\text{gen}}$, and ground truth $r^*$, the judge assigns a score on a 1–5 scale based on three key dimensions: opinion consistency, logical coherence, and stylistic fidelity. The scoring rubric emphasizes faithful persona replication, with higher scores awarded to responses that demonstrate comprehensive alignment across all dimensions.

**Ranking-based Evaluation.** The judge identifies the most persona-consistent response from a candidate set containing the generated response and the same distractors used in discriminative evaluation. This approach mirrors discriminative evaluation while leveraging the judge's nuanced understanding of persona consistency.

The generative evaluation score is computed as:

$$\text{Score}_{\text{gen}} = \frac{1}{N} \sum_{i=1}^{N} \text{Judge}(r_{\text{gen},i}, r_i^*, s_i), \tag{4}$$

where $\text{Judge}(\cdot)$ represents either the scoring or ranking function implemented by GPT-5.

## 4 BENCHMARK CONSTRUCTION

### 4.1 DATA PRE-PROCESSING

**Social Persona.** We constructed this dataset from the PChatbot Chinese microblog corpus (Qian et al., 2021). To mitigate noise and ensure each evaluation instance is meaningful, we started with 8,045 samples and applied our PCCD (Persona-Clarity and Choice-Distinctiveness) framework. We filtered for users with rich histories (average reply length of more than 10 characters; Type-Token Ratio not in the bottom 20th percentile) and for tasks with unambiguous choices (response option cosine similarity less than 0.95). We then ranked the remaining samples by a persona-choice alignment score, calculated as the similarity to the true response minus the similarity to the most similar distractor, to select the final 2,000 high-quality instances.

**Interpersonal Persona.** We used the Pushshift Telegram corpus (Baumgartner et al., 2020) to evaluate memory-grounded consistency. Our curation process followed a multi-stage filtering funnel to distill a high-quality message set from 438,975 raw messages. We first selected high-activity users (active in three or more channels with 500 or more total messages and 100 or more per channel). We then processed their messages by removing short utterances of fewer than 5 tokens, retaining only the top 10% most informative instances by TF-IDF, and applying semantic deduplication (similarity threshold of 0.90), resulting in 6,150 messages. From these, we extracted 2,500 multilingual tasks (including several languages like EN, RU, ES, PT), using GPT-5 to generate challenging distractors to ensure the task tests deep persona understanding rather than superficial cue matching. We also incorporated users' cross-channel chat history as memory to test for consistency across different social contexts.

**Narrative Persona.** We selected eight novels from the Project Gutenberg corpus (Project Gutenberg, 1971–) to test the model's ability to mimic the speaking styles of the given characters. From these novels, we extracted 1,187 speech segments covering more than 50 characters. To obtain these data, we first segmented each novel into short, indexed chunks, and from each chunk we extracted at most one utterance together with its context. We then matched the speakers to the list of main characters, whose profiles contained their personality traits, goals, motivations, and utterance histories. Once we finished collecting these speeches, each accompanied by the relevant profile and context, we constructed our test dataset, which included both multiple-choice questions and open-ended generative tasks. For the former, we paired each extracted utterance with three distractor options created based on the personalities of the other main characters. For the latter, we provided the context to the model and let it generate the most plausible utterance under the given circumstances.

### 4.2 CAPABILITY DECOMPOSITION

Guided by psycholinguistic evidence that language simultaneously conveys what people say (content) and how they say it (style) (Pennebaker et al., 2003), we coarsely group persona fidelity into two complementary dimensions: *mindset coherence* and *linguistic expression*. This view is consistent with stable individual differences in language documented across psychology and linguistics and their computational operationalizations (Costa & McCrae, 1992; Biber, 1991; Stamatatos, 2009; Neuman, 2016; Li et al., 2016). We then instantiate these dimensions with **six fundamental capabilities**: mindset coherence comprises Opinion Consistency (Zaller, 1992), Memory Recall (Clark & Brennan, 1991), and Logical Reasoning (Kahneman, 2011), whereas linguistic expression comprises Lexical Fidelity (Mehl et al., 2006; Koppel et al., 2009), Persona Tone (Brown, 1987), and Syntactic Style (Biber, 1995).

Annotation follows a prompt-aligned rubric: for each instance, annotators choose exactly one primary capability and independently assess all six capabilities as true or false under strict criteria. Capabilities are non-orthogonal by design, so multiple capabilities can be true while a single primary label captures the best-fit signal. Full instructions, criteria, and prompt excerpts appear in Appendix B, with seed examples and the JSON output format for reproducibility.

Table 2: Dataset statistics across three dimensions. Each instance corresponds to a unique persona (#Users = #Instances). Avg = average; Gen = generative; Disc = discriminative. The instruction template is counted into Token counts.

| Dimension | Instances | Avg history turns | Avg prompt tokens (Disc) | Avg prompt tokens (Gen) |
|---|---|---|---|---|
| Social Persona | 2000 | 15.0 | 1371.1 | 1215.2 |
| Interpersonal Persona | 2500 | 30.0 | 1163.5 | 1139.4 |
| Narrative Persona | 1187 | 15.7 | 934.3 | 910.7 |

Table 3: **Benchmark results for Digital Twin models:** We evaluate models using three distinct metrics: **Acc. (%)** is the accuracy on the discriminative task. **Acc. (Gen) (%)** is the accuracy where a generative model's output is evaluated via multiple choice questions by a Judge. **Score (Gen)** is a pairwise comparison score against the ground truth for generative outputs by a Judge. Higher values indicate better performance. The best result and the second best result are in **Bold** and underlined, respectively.

| | Model / Tasks | Dimension 1 | | | Dimension 2 | | | Dimension 3 | | | Average | | |
|---|---|---|---|---|---|---|---|---|---|---|---|---|---|
| | | Acc. (%) | Acc. (Gen)(%) | Score (Gen) | Acc. (%) | Acc. (Gen)(%) | Score (Gen) | Acc. (%) | Acc. (Gen)(%) | Score (Gen) | Acc. (%) | Acc. (Gen)(%) | Score (Gen) |
| | GPT-3.5-Turbo | 34.9 | 26.0 | 2.57 | 41.2 | 40.1 | 1.53 | 66.3 | 46.2 | 1.98 | 47.5 | 37.4 | 2.03 |
| | Qwen2.5-14B | 36.2 | 30.1 | 2.56 | 49.6 | 42.0 | 1.56 | 60.5 | 44.6 | 1.68 | 48.8 | 38.9 | 1.93 |
| | GPT-4o-mini | 35.3 | 26.9 | 2.61 | 39.2 | 41.3 | 1.50 | 63.1 | 46.5 | 1.91 | 45.9 | 38.2 | 2.01 |
| LLM | GPT-OSS-20B | 39.1 | 24.1 | 2.39 | 63.3 | 46.0 | 1.47 | 43.9 | 48.0 | 1.77 | 48.8 | 39.4 | 1.88 |
| | DeepSeek-V3 | 42.6 | 34.1 | **2.77** | 70.0 | 52.7 | 1.51 | 81.0 | 48.6 | 1.90 | 64.5 | 45.1 | 2.06 |
| | GPT-5-Chat | 46.9 | **38.7** | 2.73 | 77.4 | **54.0** | 1.63 | 89.4 | 52.9 | **2.03** | 71.2 | **48.5** | **2.13** |
| | Claude-Sonnet-4 | **53.9** | 37.5 | 2.67 | **84.4** | 52.9 | **1.67** | **90.2** | **53.4** | 2.02 | **76.2** | 47.9 | 2.12 |

# 5 EXPERIMENTS

## 5.1 OVERALL RESULTS AND KEY FINDINGS

We evaluate digital twin fidelity across Social, Interpersonal, and Narrative personas in two settings: a discriminative multiple-choice task and a free form generative task. Generative outputs are evaluated by GPT-5-as-a-Judge using ranking and 1 to 5 scoring. Dataset scale and prompt budgets are in Table 2, main results in Table 3, capability trends in Figure 3, and text similarity metrics in Table 5. Strong models, notably GPT-5-Chat and Claude-Sonnet-4, lead across settings, yet free form generation remains harder than the discriminative formulation, with strengths in Lexical Fidelity and Opinion Consistency and weaknesses in Persona Tone and Memory Recall. The GPT-5 Judge shows high agreement with human annotations, and BLEU-1, METEOR, and BERT-Score provide complementary evidence. Overall, the results point to remaining gaps in persona tone realization and in recalling and using persona-specific details during generation.

## 5.2 CAPABILITY-WISE ANALYSIS

We analyze performance at the capability level within our framework and present the results in Figure 3, aggregating discriminative accuracy with the two generative Judge protocols (ranking and scoring).

Three patterns emerge. First, model ranking is broadly aligned across capabilities: systems that lead on one capability tend to lead elsewhere. Second, aggregate strengths and weaknesses are stable—models score highest on *Lexical Fidelity* and *Opinion Consistency*, and lowest on *Persona Tone* and *Memory Recall*. Third, individual models show distinct comparative advantages; for example, DeepSeek-V3 approaches GPT-5 on *Lexical Fidelity* despite trailing on others. Across capabilities, the spread between LLMs is large, showing high discriminative power of the benchmark.

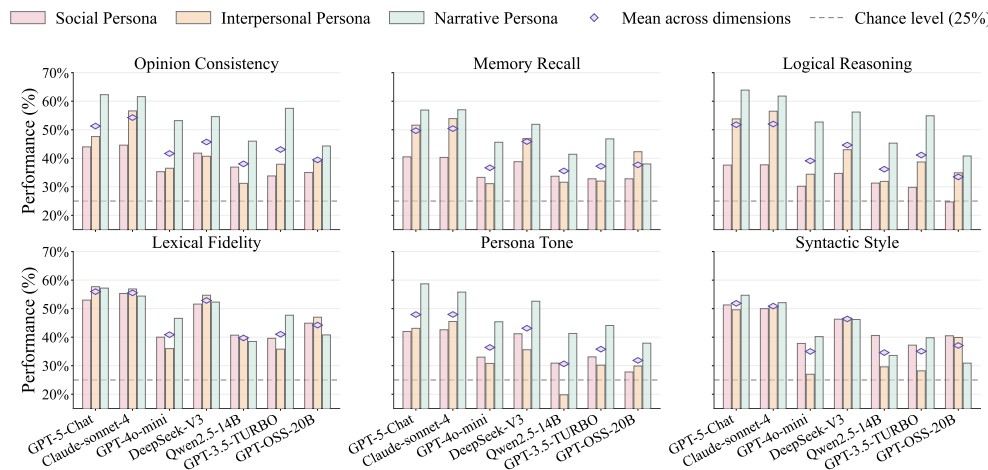

Figure 3: Performance across six capabilities. Each panel shows one capability. For each model, bars give scores on the three dimensions—Social, Interpersonal, and Narrative. Purple diamonds indicate the mean across the three dimensions for that model. The y-axis is the average over the three evaluation protocols: discriminative, generative ranking, and generative scoring. The gray dashed line denotes chance level (25%).

Table 4: Agreement of GPT-5 as a Judge against human annotations and inter-annotator reliability.

| Task | Agreement GPT-5 vs. human | Inter-annotator reliability |
|---|---|---|
| Ranking (four choice) | $0.646^{\kappa}$ | $0.673^{\kappa}$ |
| Scoring (one to five) | $0.591^{\rho}$ | $0.605^{\rho}$ |

Symbols: $\kappa$ is Cohen kappa for categorical labels and $\rho$ is Spearman correlation for ordinal scores. Sample size is 50.

## 5.3 GENERATIVE EVALUATION

### 5.3.1 LLM-AS-A-JUDGE: SCORING AND RANKING

We assess generative outputs with two Judge protocols introduced earlier, scoring (from 1 to 5) and ranking, and we aggregate their outcomes as Acc.(Gen) and Score(Gen). Full results appear in Table 3, with prompt templates and rubrics in Appendix A.

Key results are as follows: GPT-5-Chat attains the strongest aggregate generative performance (Acc.(Gen) 48.5%, Score(Gen) 2.13), closely followed by Claude-Sonnet-4 (47.9%, 2.12). DeepSeek-V3 is competitive and achieves the best Score(Gen) on the Social Persona dimension (2.77), despite trailing the leaders on other dimensions. Compared with discriminative evaluation, generative performance is systematically lower across models, underscoring the added difficulty of free-form persona simulation and the substantial headroom for improvement.

### 5.3.2 RELIABILITY OF THE JUDGE AND HUMAN STUDY

We validate the LLM-as-a-Judge methodology with a human study. Three expert annotators evaluated a stratified sample of 50 items per judging mode (ranking and scoring), following our instruction set (Appendix E). Annotators worked independently and were blinded to each other's labels.

Agreement between GPT-5-as-a-Judge and humans is reported in Table 4 and is comparable to human inter-annotator reliability: for ranking, Cohen's $\kappa$ is 0.646 (GPT-5 vs. human) versus 0.673 (human–human); for scoring, Spearman's $\rho$ is 0.591 (GPT-5 vs. human) versus 0.605 (human–human). These results indicate that the Judge is reliable, while the human inter-annotator agreement supports the quality and consistency of our annotation protocol.

Table 5: Objective metrics for Digital Twin models. We evaluate the generative outputs against the ground truth using three distinct metrics. **BLEU-1** ↑ measures unigram precision. **METEOR** ↑ considers precision, recall, and synonymy. **BERT-Score** ↑ measures semantic similarity using contextual embeddings. Higher values are better for all metrics. **Bold** numbers denote the best result and underlined numbers denote the second best in each column.

| Model / Tasks | | Dimension 1 | | | Dimension 2 | | | Dimension 3 | | | Average | |
| | BLEU-1 ↑ | METEOR ↑ | BERT-Score ↑ | BLEU-1 ↑ | METEOR ↑ | BERT-Score ↑ | BLEU-1 ↑ | METEOR ↑ | BERT-Score ↑ | BLEU-1 ↑ | METEOR ↑ | BERT-Score ↑ |
|---|---|---|---|---|---|---|---|---|---|---|---|---|
| GPT-3.5-Turbo | 16.03 | 15.50 | 62.96 | 24.76 | 22.52 | 81.54 | 12.06 | 12.86 | 84.10 | 17.62 | 16.96 | 76.20 |
| Qwen2.5-14B | 17.68 | 15.38 | 63.25 | 26.09 | 23.76 | 81.57 | 11.67 | 11.92 | 83.99 | 18.48 | 17.02 | 76.27 |
| GPT-4o-mini | 15.94 | 15.19 | 62.89 | 23.48 | 21.38 | 81.26 | **12.50** | **13.34** | 84.13 | 17.31 | 16.64 | 76.09 |
| GPT-OSS-20B | 14.55 | 12.87 | 61.90 | 20.67 | 19.20 | 81.17 | 10.81 | 10.59 | 84.36 | 15.34 | 14.22 | 75.81 |
| DeepSeek-V3 | 16.85 | 15.49 | 63.25 | 26.86 | 25.21 | 82.65 | 11.11 | 11.58 | 84.12 | 18.27 | 17.43 | 76.67 |
| GPT-5-Chat | 18.67 | 14.09 | 63.26 | **27.18** | **25.30** | **82.67** | 11.54 | 11.59 | 84.27 | **19.13** | 16.99 | 76.73 |
| Claude-Sonnet-4 | **18.68** | **18.14** | **64.19** | 25.22 | 23.45 | 82.14 | 12.38 | 13.12 | **84.37** | 18.76 | **18.24** | **76.90** |

(LLM)

Table 6: Discriminative evaluation against a reference standard

| Task | Accuracy | | | Agreement ($\kappa$) | |
| | GPT-5 | Human mean | Human vote | Model vs human | Inter-annotator |
|---|---|---|---|---|---|
| Discriminative | 0.60 | 0.64 | 0.66 | 0.634 | 0.690 |

Human mean is the average across individual annotators. Majority vote accuracy evaluates the aggregated vote by annotators. Agreement uses Cohen kappa $\kappa$. Sample size is 50.

### 5.3.3 TEXT SIMILARITY METRICS

To provide an objective reference, we also evaluate free-form generations with standard text similarity metrics—BLEU-1, METEOR, and BERT-Score—and report results in Table 5. Averaged over the three dimensions, Claude-Sonnet-4 attains the best BERT-Score (76.90) and METEOR (18.24), while GPT-5-Chat achieves the best BLEU-1 (19.13). The resulting model ranking is broadly consistent with our judge-based evaluation, offering cross-validation. These metrics primarily reflect lexical overlap and local paraphrase rather than opinion alignment, reasoning trajectories, or persona tone. Therefore, we treat them as complementary evidence to judge-based results.

### 5.4 HUMAN VS. MODEL PERFORMANCE

We benchmark human performance on the Social Persona discriminative task. Three expert annotators labeled a stratified set of 50 items following our guidelines (Appendix E). Because persona simulation involves long contexts and implicit cues, we do not treat human accuracy as a strict upper bound.

Table 6 compares models to human baselines. GPT-5-Chat reaches 0.60 accuracy, below the human mean of 0.64 and the majority-vote aggregate of 0.66. Agreement with humans is high but short of human–human reliability: Cohen's $\kappa$ is 0.634 for model vs. human and 0.690 for inter-annotator agreement.

These results indicate that state-of-the-art models approach human reliability on this discriminative formulation yet still trail aggregated human judgments, leaving measurable headroom. Given that humans are imperfect simulators in this setting, we view these numbers as practical reference points rather than hard ceilings.

**Summary of Findings.** Across three persona dimensions and two task formulations, strong models (GPT-5-Chat, Claude-Sonnet-4) lead consistently, yet free-form persona simulation remains notably harder than multiple-choice selection. Capability analysis pinpoints style control and memory recall as primary bottlenecks, while lexical fidelity and opinion consistency are comparatively robust. GPT-5-as-a-Judge provides reliable, scalable assessment that aligns with human judgments, and text-similarity metrics offer complementary confirmation. Across settings, results exhibit substantial variance between models without evident ceiling effects. There remains clear headroom in three areas: maintaining persona coherence over extended contexts and across sessions, producing a persona-consistent tone, and recalling and using persona-specific facts during generation.

# 6 CONCLUSIONS AND DISCUSSIONS

This paper addressed the evaluation of LLM-based persona simulation by introducing **TwinVoice**. Built on real-world and fictional data from three dimensions, TwinVoice aims at testing LLMs' ability in persona simulation by decomposing it into six capabilities of mindset coherence and linguistic expression. Our extensive evaluation of state-of-the-art models reveals a crucial gap: while leading models like GPT-5-Chat and Claude-Sonnet-4 show improved accuracy over their predecessors, their performance still falls significantly short of human-level capabilities. We also find that LLMs are adept at mimicking surface-level linguistic styles, they consistently fail to maintain long-term consistency, particularly in memory recall and opinion stability. By establishing the first fine-grained baselines in this domain, TwinVoice not only exposes the key limitations of current models but also provides a clear roadmap towards personalized AI and digital twins built with LLMs.

**Rationale for Three Dimensions.** TwinVoice is constructed based on Social, Interpersonal, and Narrative personas to balance realism, coverage, and privacy. Social and Interpersonal tracks are built on real interaction traces because evaluating digital twins requires performance in authentic public and private contexts; synthetic or model-generated corpora alone underestimate the difficulty of sustaining identity over long horizons. For Narrative persona, full real-world narrative streams are hard to obtain and raise privacy concerns; we therefore use curated fiction to probe role-consistent expression under controlled, ethically tractable settings.

**Evaluation Design.** Digital twins must go beyond constrained selection to produce persona-consistent language under open prompts. We therefore pair a discriminative multiple-choice protocol (with carefully constructed, topically plausible distractors) with a generative protocol that assesses free-form responses using two LLM-as-a-Judge variants (ranking and scoring) along opinion consistency, logical/factual fidelity, and stylistic similarity. Judge reliability is supported by a human study with three expert annotators: GPT-5-as-a-Judge reaches agreement close to human inter-annotator levels (ranking $\kappa \approx 0.646$ vs. $0.673$; scoring $\rho \approx 0.591$ vs. $0.605$).

**Usability, Reproducibility, and Robustness.** We release precise task definitions, prompts, and data paths so researchers can plug in fine-tuning, RAG, long-term memory, or multi-agent controllers on the same inputs. For generation, we fix `temperature`=0.0 and publish decoding settings, seeds, and candidate-construction scripts; we log model build identifiers where available and release raw outputs to mitigate closed-API drift. Social Persona derives from PChatbot; to reduce leakage we enforce semantic distinctiveness in choice sets and apply persona–choice alignment filters, and we plan annual refreshes to retire suspect items. With parallelism set to 10, end-to-end evaluation per model per dimension completes within 2 hours on our setup.

**Coverage and Limitations.** TwinVoice currently spans three dimensions and five languages: Social (Chinese), Interpersonal (English, Spanish, Portuguese, Russian), and Narrative (English). Despite this breadth, language balance within each dimension remains imperfect, and phenomena such as code-switching and dialectal variation are underrepresented. Future releases will expand per-dimension language coverage and diversify domains where consented and de-identified data are available.

**Maintenance and Outlook.** We will maintain TwinVoice with annual updates to address potential contamination, accommodate new model behaviors, and extend language and domain coverage. Planned upgrades include longer-horizon tasks that jointly stress memory and opinion stability, adversarial tone/stance confounders for robustness, and, where ethically permissible, additional dimensions and task types. All releases will be versioned, with code and results publicly available for reproducibility.

## ETHICS STATEMENT

We follow standard ethical guidelines for dataset usage, evaluation, and model deployment. All datasets used in this paper are publicly available under their original licenses, and we removed personally identifiable information (PII) where applicable. No human subjects experiments were conducted beyond voluntary annotation; annotators (if any) received fair compensation and provided informed consent. We prohibit misuse of our benchmark and models for profiling or harmful decision making about individuals. Third-party models/APIs used in our experiments comply with their terms of service. Upon acceptance, we will release our code, prompts, and evaluation scripts with a research license and a model card detailing limitations and appropriate use.

## REPRODUCIBILITY STATEMENT

We enable independent re-implementation of our evaluation by disclosing all essential ingredients in the paper and appendices:

- **Prompts & Protocols:** Full templates for the discriminative MCQ task, generative persona imitation, and LLM-as-a-Judge (ranking and scoring), together with the 1–5 scoring rubric aligned with opinion, logic/facts, and style.
- **Data Construction Recipes:** Step-by-step textual recipes for all three dimensions, including sources and filtering thresholds (e.g., average reply length $> 10$, bottom-20% TTR removal, option cosine similarity $< 0.95$ for Social; token-length cleaning $< 5$, TF–IDF top-10% selection, and semantic deduplication at $0.90$ for Interpersonal), and the rules used to form distractors.
- **Dataset Statistics:** Per-dimension instance counts and summary statistics as reported in the main text.
- **Evaluation Definitions:** Exact metrics and equations (e.g., Accuracy and $\mathrm{Score}_{\mathrm{gen}}$) used throughout.
- **Model Usage:** The list of model families evaluated and our access window (06/2025–09/2025). We set the decoding temperature to 0 (`temperature=0`); all other generation hyperparameters (e.g., top_p, max_tokens, presence/frequency penalties) used provider defaults.

All experiments are inference-only (no supervised training). With these disclosed materials, readers can re-implement the pipeline and obtain comparable results under the same inputs and judging criteria.

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

CONTENTS

## A  EVALUATION PROTOCOLS AND FULL PROMPTS

This appendix details our evaluation protocols and the full instruction templates used across multiple data forms, including public social interactions, interpersonal messaging, and narrative dialogue. We adopt a unified instruction design and provide template variants for different data shapes when needed. Unless otherwise noted, the LLM-as-a-Judge component is instantiated with GPT-5.

### A.1  SCOPE AND ALIGNMENT WITH COMPETENCIES

Our evaluation comprises (1) discriminative multiple-choice selection and (2) generative evaluation, including persona imitation (free-form generation) and LLM-as-a-Judge assessment via ranking and scoring. The judge scoring rubric is organized along three pillars—Opinion Consistency, Logical & Factual Fidelity, and Stylistic Similarity—which align with the six fundamental capabilities defined in the main text. We offer equivalent template variants per evaluation mode to fit different data shapes; metrics and scoring criteria remain identical across variants.

### A.2  UNIFYING INSTRUCTIONS AND PLACEHOLDERS

We use a single instruction family per evaluation mode. Differences are limited to how inputs are presented. We standardize placeholders as follows:

- {history}: persona-establishing prior content by the same user or character.
- {context}: the situation/post/message/scene the user or character is responding to (replacing earlier {anchor} or {anchor_post}).
- {ground_truth_reply} or {groundtruth_response}: the human-written reply.
- {lmut_reply} or {generated_content}: the model-generated reply to be evaluated.

### A.3  DISCRIMINATIVE EVALUATION (MULTIPLE-CHOICE SELECTION)

**Canonical template (General).**

---
**Discriminative Selection Prompt (General)**

```
Your task is to act as a specific social media user, becoming their
    digital twin.
Note: All provided text (history, context, choices) is in the
    original language of the data. You must analyze the user's style
    directly within that language.

Based on the user's reply history, think and respond with their
    mindset, tone, and style.

Your reply history:
(Note: ``Context'' is another user's post/message, and ``UserReply
    '' is your own reply.)
{history}

Now, you see a new context message:
``{context}''

Below are 4 candidate replies. Which one is most likely something
    you would say?

A. {a}
B. {b}
C. {c}
D. {d}
```

---

```
Please respond in the following JSON format. In the ``reasoning''
    field, use the first-person perspective (``I'') to explain your
    choice.

```json
{{
  "predicted_comment": "A",
  "reasoning": "Explain, from my perspective as the user, why I
      would choose this option."
}}
```
```

**Alternative template (Dimension 2: Interpersonal Messaging).**

**Discriminative Selection Prompt (Messaging Variant)**

```
You are given a user's reply history and 4 candidate replies to a
    context message. Only one of the replies was actually written by
     this user. The other three were written by different users
    replying to the same context message.
Your task is to choose the most likely reply written by the same
    user, based on writing style, tone, and expression habits. Focus
     on how the user typically speaks, their phrasing, and how they
    respond emotionally or humorously.

User's Historical Conversations:
{history}

Current Context Message:
``{context}''

Candidate Replies:
A. {a}
B. {b}
C. {c}
D. {d}

Please respond in the following JSON format:
```json
{{
  "predicted_comment": "A",
  "reasoning": "Explain why this option best matches the user's
      style."
}}
```
```

**Distractor Generation for Discriminative Data (Dimension 3: Narrative).**

**Distractor Writer Prompt (Narrative Variant)**

```
You are a precise persona-grounded writer.
Given one TARGET speaker (whose original utterance is the correct
    answer) and THREE OTHER characters, write EXACTLY THREE
    distractor lines that those other characters would plausibly say
     in this context.

Return ONLY this JSON:
{{
  "distractors":[
```

```
      {{"text":"...", "by":"<OtherCharacterName>"}},
      {{"text":"...", "by":"<OtherCharacterName>"}},
      {{"text":"...", "by":"<OtherCharacterName>"}}
    ]
  }}

  Context (narration BEFORE anyone speaks):
  """{context_text}"""

  TARGET (do NOT imitate in distractors):
  - name: {target_name}
  - traits: {t_traits}
  - goals: {t_goals}
  - details: {t_details}
  - history: {t_history}

  THREE OTHER characters (write one distractor for each; must sound
      like them):
  1) name: {o1_name}
     traits: {o1_traits}
     goals: {o1_goals}
     details: {o1_details}
     history: {o1_history}
  2) name: {o2_name}
     traits: {o2_traits}
     goals: {o2_goals}
     details: {o2_details}
     history: {o2_history}
  3) name: {o3_name}
     traits: {o3_traits}
     goals: {o3_goals}
     details: {o3_details}
     history: {o3_history}

  Rules (STRICT):
  - Context fit: Each distractor must be logically possible GIVEN the
       context (time/place/people/danger level). Do NOT introduce
      facts that contradict the context (e.g., saying ``it's calm''
      when the scene is a chase or fire).
  - Persona fit: Each distractor must match the specified OTHER
      character's traits/goals/details AND be consistent with their
      history. Do NOT copy, paraphrase, or stylistically mimic the
      TARGET.
  - History use: Use the OTHER character's UtteranceHistory to guide
      tone, stance, formality, and typical verbs; NEVER copy any
      sentence from history verbatim. Avoid the TARGET's pet phrases
      or signature moves.
  - Style \& length: Keep 1 short line per distractor, in the book's
      tone/era (no modern slang/emojis). Prefer 8--28 words;
      comparable length to a typical line in this book. Natural
      punctuation (commas/semicolons/em dashes) is OK.
  - Voice: No stage directions, no ``X said,'' no speaker names in
      the line. The content should read as the spoken line itself.
  - Uniqueness: The three distractors must be meaningfully different
      in stance/wording; no near-duplicates.
  - Safety checks:
   * If any distractor contradicts the context, resembles the TARGET'
      s voice, copies history verbatim, or breaks style/length
      constraints, REWRITE it.
   * Output EXACTLY three items; no extra keys or commentary.

  Output ONLY the JSON object described above.
```

**Notes.**

- Placeholders are standardized: $\{\texttt{history}\}$, $\{\texttt{context}\}$, and option texts $\{\texttt{a}\}$, $\{\texttt{b}\}$, $\{\texttt{c}\}$, $\{\texttt{d}\}$. In narrative data, the distractor writer prompt is used to construct options and is not itself a judging template.

## A.4  GENERATIVE EVALUATION: PERSONA IMITATION (FREE-FORM GENERATION)

**Canonical template (General, text-only output).**

```
Generative Persona Imitation Prompt (General)

You are acting as a digital twin of a specific social media user.
Your task is to analyze the user's posting history to understand
    their personality, tone, vocabulary, and style.
All provided text (history, context) is in the original language of
    the data. You must analyze and respond in that language.

Here is the user's posting history:
(Note: ''Context'' is a post/message by someone else, and ''
    UserReply'' is the user's own reply to it.)
---
{history_text}
---

Now, you must imitate this user's persona perfectly and write a new
    reply to the following message.
Respond ONLY with the text of the reply. Do not add any extra
    explanations, greetings, or surrounding text.

Message to reply to:
''{context}''
```

**Variant (Dimension 2: Messaging, JSON output).**

```
LMUT Prompt (Messaging Variant, JSON Output)

You are acting as a digital twin of a specific messaging app user.
Your task is to analyze the user's messaging history to understand
    their personality, tone, vocabulary, and style.
Different provided text (history, context, message) may use
    different language. You must analyze and respond in the same
    language as the provided text.

Here is the user's messaging history:
(Note: ''Context'' is a message by someone else, and ''UserReply''
    is the user's own reply to it.)
---
{history_text}
---

Now, you must imitate this user's persona perfectly and write a new
    reply to the following message.
Please include your response in the following JSON format:
{{"generated_content": "your reply text here"}}
You may include thinking process or other content, but make sure to
    include the JSON format with the generated_content field.

Message to reply to:
''{context}''
```

**Variant (Dimension 3: Narrative, single-line JSON).**

**Digital Twin Line Generation (Narrative Variant)**

```
You are the digital twin of the TARGET speaker in a literary
    dialogue dataset.

Your job: write ONE new reply that this TARGET would plausibly say
    in the exact scene below, matching their historical voice and
    habits.

### Inputs
- TARGET speaker: {speaker}
- Scene context (preceding narration \& situation, NOT the speaker'
    s own words):
"""{context}"""
- (Optional) TARGET's conversation history snippets (style anchors)
    :
{history_block}

### Hard requirements (STRICT)
1) Language \& Era: Match the book's tone/era (no modern slang/
    emojis). If the scene reads like 19th-century prose, mimic that
    diction.
2) Persona Fit: Keep the TARGET's typical formality, sentence
    length, cadence, favorite turns of phrase (use hints from
    history if provided).
3) Scene Consistency: The line must be logically possible given the
    context. Do NOT introduce new facts/characters/locations. No
    meta-commentary.
4) Length \& Shape: One spoken line only (no stage directions, no
    speaker tag). Prefer 8--28 words unless the scene clearly calls
    for a very short assent/command.
5) No Copying: Do NOT copy any exact sentence from the dataset.
    Paraphrase in the TARGET's voice.
6) Output format: Return ONLY a JSON object:
{{
  "generated_content": "<the single line>"
}}

Now produce the JSON with your single-line reply.
```

**Notes.**

- Use {context} as the reply trigger across all variants. The narrative variant mandates a single-line JSON output.

## A.5 LLM-AS-A-JUDGE: RANKING-BASED EVALUATION

**Canonical template (JSON + concise reasoning).**

---

**Judge Ranking Prompt (General)**

```
You are an expert evaluator of writing style. Your task is to
    compare several candidate replies against a known ``Reference
    Reply'' written by a specific user.

Your goal is to identify which candidate is the most similar to the
    reference in terms of **style, tone, vocabulary, sentiment, and
    topic**.

This is the Reference Reply (the ground truth written by the user):
---
{ground_truth_reply}
---

These are the **Candidate Replies**:
{candidate_replies_text}

Now, determine which single candidate is the closest match to the
    Reference Reply.
You MUST respond ONLY with a JSON object in the following format.
    Do not include any other text.
The reasoning should be concise, limited to 2--3 sentences.

```json
{{
  "choice": "The letter of the best matching candidate (e.g., 'A', '
      B', 'C', or 'D')",
  "reasoning": "A brief explanation for your choice, focusing on the
      stylistic similarities."
}}
```
```

---

**Letter-only MAP Prompt (Dimension 3: Narrative).**

---

**MAP Prompt (Narrative Variant, Letter Only)**

```
You are a strict classifier. Output ONLY a single letter (A/B/C/D).
Choose the option that best matches the style, tone, vocabulary,
    and stance of the Generated Reply.

[Options]
A. {A}
B. {B}
C. {C}
D. {D}

[Generated Reply]
{pred}

Output exactly one letter: A, B, C, or D.
```

---

**Notes.**

- Ranking supports two outputs: a JSON object with brief reasoning (general) and a letter-only output (narrative variant).

## A.6 LLM-AS-A-JUDGE: SCORING-BASED EVALUATION

**Canonical template (applies as-is).**

> **Judge Scoring Prompt (All Variants)**
>
> ```
> You are a meticulous and objective evaluator for a digital twin
>     benchmark. Your task is to assess how well a ``Generated Reply''
>      replicates a ``Ground Truth Reply'' for a given interaction.
>
> The ``Ground Truth Reply'' is the single, undisputed gold standard.
>      Your entire evaluation must be based on comparing the ``
>     Generated Reply'' against it.
>
> The evaluation rests on three key pillars:
> 1. **Opinion Consistency**: Does the ``Generated Reply'' express
>     the exact same core opinion, stance, and sentiment as the ``
>     Ground Truth''?
> 2. **Logical \& Factual Fidelity**: Is the ``Generated Reply''
>     based on the same reasoning and facts as the ``Ground Truth''?
>      It must not introduce new, unsupported information or follow a
>      different logical path.
> 3. **Stylistic Similarity**: How closely does the ``Generated Reply
>     '' match the ``Ground Truth'' in terms of writing style?
>     * **Lexical**: Use of similar vocabulary, slang, or emojis.
>     * **Tone**: Capturing the same tone (e.g., humorous, sarcastic,
>         empathetic, proud).
>     * **Syntactic**: Similarity in sentence structure, length, and
>         degree of formality.
>
> ---
> SCORING RUBRIC (1--5 Scale):
>
> - **5: Perfect Replication**: The ``Generated Reply'' is a perfect
>     match across all three pillars (Opinion, Logic/Factual, Style).
>      It feels like a natural, alternative expression from the same
>     user. A perfect substitute for the ground truth.
>
> - **4: High Fidelity**: The Opinion and Logic/Factual pillars are
>     perfectly matched. There are only minor, subtle differences in
>     the Style pillar (e.g., a missing catchphrase, a slightly more
>     formal tone), but the reply is still an excellent imitation.
>
> - **3: Core Alignment, Detail Loss**: The core Opinion is
>     consistent, but there's a noticeable loss of detail in the Logic
>      or Style pillars. For example, the tone is flattened, or unique
>      phrasing is lost. The reply captures the ``what'' but not the
>     ``how''. It feels more like a summary than a replication.
>
> - **2: Partial Relevance, Major Deviation**: There is a major
>     failure in at least one of the three pillars. For instance, the
>     opinion is distorted (e.g., strong support becomes neutral), the
>      logic is completely different, or the style is entirely
>     mismatched.
>
> - **1: Irrelevant or Contradictory**: The ``Generated Reply'' has
>     almost nothing in common with the ``Ground Truth'' or expresses
>     a contradictory opinion. A total failure of replication.
>
> ---
> YOUR TASK:
> You will be provided with the context message, the ground truth
>     reply, and the generated reply. User-generated content may be in
> ```

```
     different languages, but your analysis and final JSON output
     must be in English. You MUST respond ONLY with a JSON object in
     the following format. Do not include any other text or
     explanations.

```json
{{
  "analysis": {{
    "opinion_consistency": {{
      "is_consistent": true,
      "justification": "A brief justification for the consistency of
          the opinion."
    }},
    "logical_factual_fidelity": {{
      "is_faithful": true,
      "justification": "A brief justification for the fidelity of the
          logic and facts."
    }},
    "stylistic_similarity": {{
      "similarity_level": "High/Medium/Low",
      "justification": "A brief justification for the level of
          stylistic similarity."
    }}
  }},
  "final_score": "An integer score from 1 to 5",
  "final_justification": "A concise, overall justification for the
      final score, synthesizing the three pillars."
}}
Now, evaluate the following case:

Context Message:
``{context}``

Ground Truth Reply:
``{ground_truth_reply}``

Generated Reply to Evaluate:
``{lmut_reply}``
```

**Notes.**

- Inputs are standardized as {history}, {context}, {ground_truth_reply} (or {groundtruth_response}), and {lmut_reply} (or {generated_content}).

## A.7 IMPLEMENTATION NOTE: JUDGE MODEL

We instantiate the LLM-as-a-Judge with GPT-5 for both ranking- and scoring-based evaluation, unless otherwise specified. Ranking includes a letter-only variant for narrative data.

## B  CAPABILITY ANNOTATION PROMPTS AND LABELING PROTOCOL

We annotate each example to identify which capability a model must primarily exercise to replicate a user's reply, while also recording the presence of all six capabilities. Each annotation unit contains three elements: {history} (persona-establishing prior content), {context} (the situation the user is replying to), and {groundtruth_response} (the user's actual reply). An expert LLM performs the annotation to ensure consistency and structured outputs (we use GPT-5 with temperature set to 0).

**Canonical Annotation Prompt.**

---

**Capability Annotation Prompt (Canonical)**

```
\# ROLE AND GOAL
You are an expert linguistic and persona analyst. Your task is to
    analyze user data to identify the core capabilities a generative
     model would need to successfully create a ``digital twin'' of
    the user. You will be given a user's conversational history, a
    new context they are replying to, and their actual response (``
    groundtruth'').

\# INPUT DATA STRUCTURE
You will receive a JSON object for each annotation task with the
    following structure:
- ``context'': The situation, post, or utterance the user is
    responding to.
- ``groundtruth\_response'': The user's actual, human-written
    response to the ``context''.
- ``history'': A list of the user's past posts and replies, which
    establishes their persona.

\# CORE TASK: CAPABILITY ANNOTATION
Your task is twofold.
Part 1 is mandatory: You must first identify the single ``primary\
    _capability''. This is the one capability that serves as the
    best-fit or most representative label for the example, even if
    the signal is weak. This choice is required for every single
    data point.
Part 2 is for detail: After identifying the primary capability, you
     will then perform a standard evaluation for all six
    capabilities, marking ``true'' or ``false'' for each based on
    the strict criteria. This allows for multiple capabilities to be
     ``true''.

\# CAPABILITY DEFINITIONS AND ANNOTATION CRITERIA
Evaluate each capability independently based on the refined
    criteria below.

C1: Opinion Consistency
- Core Question: Does this response require explicitly reaffirming
    a specific, previously-stated opinion?
- Label ``true'' if: The ``groundtruth\_response'' expresses a
    clear opinion (e.g., support for a team, dislike for a policy)
    that directly and unambiguously repeats or reinforces an opinion
     explicitly stated in the ``history''.
- Do not label ``true'' for new opinions on new topics, even if
    they seem plausible for the user, or for generic positive/
    negative sentiment that isn't tied to a specific, recurring
    viewpoint.
- Choose as ``primary\_capability'' if: The core purpose of the
    response is to state a known, consistent opinion.
```

---

```
C2: Memory\_Recall
- Core Question: Does the response rely on shared context or
    information from the history that an outside reader would not
    fully understand?
- Label ``true'' if: The ``groundtruth\_response'' makes an
    explicit or implicit reference to a past event, personal
    information, or a previously established piece of context from
    the ``history''.
- Do not label ``true'': If the response is entirely self-contained
     and can be perfectly understood by anyone just by reading the
    ``context''.
- Choose as ``primary\_capability'' if: The response would be
    confusing or lose its meaning without knowledge of the user's
    history. This is often a good default choice for very short,
    context-dependent replies.

C3: Logical Reasoning
- Core Question: Does this response provide a justification or
    explanation for a claim?
- Label ``true'' if: The ``groundtruth\_response'' contains a
    rationale (e.g., using ``because,'' ``since,'' ``so,'' or
    implying a cause-and-effect relationship), AND the user's ``
    history'' shows a pattern of them providing reasons for their
    opinions.
- Do not label ``true'': If the response is a simple, unsupported
    statement of fact or feeling.
- Choose as ``primary\_capability'' if: The response structure is
    clearly ``Claim + Justification''.

C4: Lexical\_Fidelity
- Core Question: Does this response use a creative, personal, and
    repeated signature word or phrase?
- Label ``true'' if: The ``groundtruth\_response'' uses a specific
    word, phrase, or emoji pattern that is both repeated in the ``
    history'' and idiosyncratic (not common slang).
- Do not label ``true'': For common slang or any single-use clever
    phrase.
- Choose as ``primary\_capability'' if: The most noticeable feature
     of the response is the use of a signature word/phrase.

C5: Persona\_Tone
- Core Question: Does the response use a specific, non-literal tone
     (like sarcasm or deep irony) that is a core part of the user's
    persona?
- Label ``true'' only if: The history shows a recurring pattern of
    a specific, non-literal tone AND the response is a clear
    instance of that same tone.
- Do not label ``true'': If the two strict conditions are not both
    met.
- Choose as ``primary\_capability'' if: The meaning of the response
     is inverted or altered by a clear, persona-defining tone (e.g.,
     obvious sarcasm).

C6: Syntactic\_Style
- Core Question: Does this response use a distinctive, repeated
    structural pattern?
- Label ``true'' only if: The response uses a clear, repeated, and
    non-standard stylistic pattern (e.g., habitual use of sentence
    fragments, a unique punctuation signature).
- Do not label ``true'': For common conversational variations.
- Choose as ``primary\_capability'' if: The response is very simple
    and its most defining characteristic is a structural quirk (e.g
    ., it's just a single, fragmented phrase, which is a common
```

```
      pattern for the user). This can be a fallback for otherwise
      simple responses.

  \# INSTRUCTIONS \& OUTPUT FORMAT
  1. Step 1: Determine the ``primary\_capability'' (Mandatory Choice)
      .
      - First, analyze all the data.
      - To ensure a fair evaluation and eliminate any potential
          ordering bias, you must give equal and independent
          consideration to all six capabilities, regardless of their
          order, before selecting the primary\_capability.
      - Then, you MUST choose exactly one capability from the list (C1
          --C6) that you consider the best fit.
      - Use the ``Choose as primary\_capability if...'' guidelines to
          help you decide. If no signal is strong, choose the one that
          is the most plausible or least incorrect. For very generic
          replies, ``Memory\_Recall'' or ``Syntactic\_Style'' are often
           good candidates.
      - This choice is not optional.

  2. Step 2: Evaluate All Capabilities (Detailed Annotation).
      - Now, go through each of the six capabilities (C1 to C6) one by
          one, including the one you chose as primary.
      - For each one, decide if the ``groundtruth\_response'' meets
          the strict definition and assign ``true'' or ``false''.
      - Provide a brief, one-sentence justification for every
          capability you mark as ``true''.

  3. Step 3: Format the Output.
      - Your final output must be a single, valid JSON object with the
          exact two-level structure shown below.
      - The ``primary\_capability'' field MUST contain the string name
          of your choice from Step 1. It cannot be null or empty.
      - The ``all\_evaluations'' field MUST contain the detailed
          boolean labels from Step 2.

  ```json
  {
    "primary_capability": "Name_Of_The_Single_Best_Fit_Capability",
    "all_evaluations": {
      "Opinion_Consistency": { "label": false, "reasoning": "" },
      "Memory_Recall": { "label": false, "reasoning": "" },
      "Logical_Reasoning": { "label": false, "reasoning": "" },
      "Lexical_Fidelity": { "label": false, "reasoning": "" },
      "Persona_Tone": { "label": false, "reasoning": "" },
      "Syntactic_Style": { "label": false, "reasoning": "" }
    }
  }
  ```
```

**Inputs for the prompt.**   We pass a single JSON object per example with three keys: `history`, `context`, and `groundtruth_response`. No length truncation is applied.

# C  CAPABILITY DISTINGUISHING CASE STUDIES

This section presents case studies that illustrate how our six capabilities appear in practice. The examples are drawn from our public social persona corpus (dimension 1). For readability we show faithful translations and only the key slices. If any discrepancy arises, the original Chinese dataset is authoritative. Explanatory remarks appear outside the boxes. Inside each box, ······ marks omitted portions of longer cases.

## C.1  OPINION CONSISTENCY

The user maintains a specific stance across contexts, namely choosing shows based on a favorite actor and praising acting skill. The new reply preserves this granular stance rather than defaulting to generic positivity.

---

**Case 1: Opinion Consistency (user 527222)**

**Context.** ······ "Tonight is the finale. Xiang Qian returns to the seaside house where he once lived in hard times, surely full of feelings. Seeing Alisa in this moment is so beautiful, hope they both have a good life." ······
**Key History.** ······ "I watched this show for Huang Zitao, I think his acting is great." ······
**Ground Truth Reply.** ······ "I watched it for Liu Tao, her acting is really getting better and better." ······

---

Why this shows Opinion Consistency: The historical pattern is watch for a specific actor and praise that acting. The ground truth reply mirrors the same stance toward another named actor, preserving topic granularity and evaluative angle.

## C.2  MEMORY RECALL

The reply uses a nickname that is not introduced in the immediate context, presupposing shared knowledge from prior interactions. Understanding the line fully requires recalling who that nickname refers to.

---

**Case 2: Memory Recall (user 205470)**

**Context.** ······ "Met a teacher who is a high level LEGO player, buys LEGO by the sack." ······
**Key History.** ······ "When Dan jie builds LEGO she looks like a serious kid, always supporting Dan jie." ······
**Ground Truth Reply.** ······ "When she plays LEGO her eyes light up, still that adorable Wang Sansui." ······

---

Why this shows Memory Recall: The affectionate nickname Wang Sansui is not grounded in the current context and relies on earlier persona knowledge to resolve the reference.

## C.3 LOGICAL REASONING

The user's pattern is Observation then Deduction. In history, a physical observation supports an inference. The reply replicates this approach by citing scene features to argue against an assumption.

---

**Case 3: Logical Reasoning (user 369593)**

**Context.** ······ "Do an ice drifting video. If it is not minus twenty or thirty degrees, do not show off." ······
**Key History.** ······ "There is no snow on the roof opposite, which shows the heat inside that house is considerable." ······
**Ground Truth Reply.** ······ "This river channel is quite narrow and there is a road next to it, so it probably did not fall in from drifting on the ice." ······

---

Why this shows Logical Reasoning: The reply marshals concrete observations (narrow channel, road nearby) to support a causal judgment, matching the user's habit of evidence based inference.

## C.4 LEXICAL FIDELITY

A personal catchphrase recurs across contexts. The reply deploys the same idiosyncratic exclamation seen in history, signaling a learned lexical signature.

---

**Case 4: Lexical Fidelity (user 45899)**

**Context.** ······ "Emirates Bling777 plane is encrusted with Swarovski crystals, the joy of the rich is beyond imagination." ······
**Key History.** ······ "OMG, for this kind of dog, give me a dozen and it is not too many." ······
**Ground Truth Reply.** ······ "OMG, this, this, it is full of diamonds?! Maybe one will drop off for me." ······

---

Why this shows Lexical Fidelity: The same colloquial exclamation equivalent to OMG appears in both history and reply, demonstrating consistent, user specific lexical choice.

C.5 PERSONA TONE

The user favors playful hyperbole and adoring expressions that are nonliteral. The reply echoes that tone with a different bodily metaphor, preserving the same stylistic stance.

---

**Case 5: Persona Tone (user 270844)**

**Context.** ······ "Group stage, Hai Lu's acting is on point, those long legs are eye catching. Did not expect such solid dance foundation, the high kicks are captivating." ······
**Key History.** ······ "Listening made my ears pregnant, you all should listen, it is super good. Hope my male god keeps getting better. Could you be my boyfriend, so shy." ······
**Ground Truth Reply.** ······ "Hai Lu, your long legs had me staring at them the whole time, haha, my nose is about to bleed." ······

---

Why this shows Persona Tone: Both history and reply use exuberant, nonliteral bodily metaphors (ears pregnant, nosebleed) as playful, adoring exaggerations that define the user's persona.

C.6 SYNTACTIC STYLE

Beyond words and tone, the user's structure features stacked, breathless exclamations with intensifiers. The reply reproduces that sentence shape.

---

**Case 6: Syntactic Style (user 108194)**

**Context.** ······ "Sci fi fans, gather up. The film The Wandering Earth is set for Lunar New Year, a concept poster has been released." ······
**Key History.** ······ "Wow wow wow, I am truly so excited inside, really looking forward to it, hahaha." ······
**Ground Truth Reply.** ······ "Wow wow wow, look closely, this poster design really has such a vibe, you could call it outstanding. This kind of movie theme is especially attractive, must support." ······

---

Why this shows Syntactic Style: The reply stacks short, exclamatory clauses with intensifiers and colloquial particles, recreating the user's distinctive, breathless rhythm observed in history.

## D   RADAR CHARTS ACROSS THREE DIMENSIONS

We present capability-wise radar charts for the three persona dimensions: Dimension 1 (Social Persona), Dimension 2 (Interpersonal Persona), and Dimension 3 (Narrative Persona). For each dimension, we report four evaluation configurations: (i) Combined Average (aggregated across protocols), (ii) Discriminative (multiple-choice selection), (iii) Generative Ranking (LLM-as-a-Judge; Acc.(Gen)), and (iv) Generative Scoring (LLM-as-a-Judge; Score(Gen), 1–5). Each radar covers six capabilities: Opinion Consistency, Memory Recall, Logical Reasoning, Lexical Fidelity, Persona Tone, and Syntactic Style.

### D.1   SOCIAL PERSONA (DIMENSION 1)

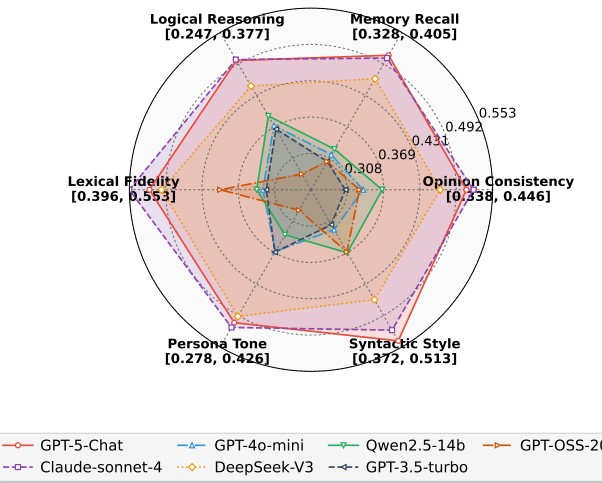

Figure 4: Dimension 1 (Social Persona): Combined Average radar over six capabilities (all labeled capabilities). Aggregates across discriminative and generative protocols; higher is better along each spoke.

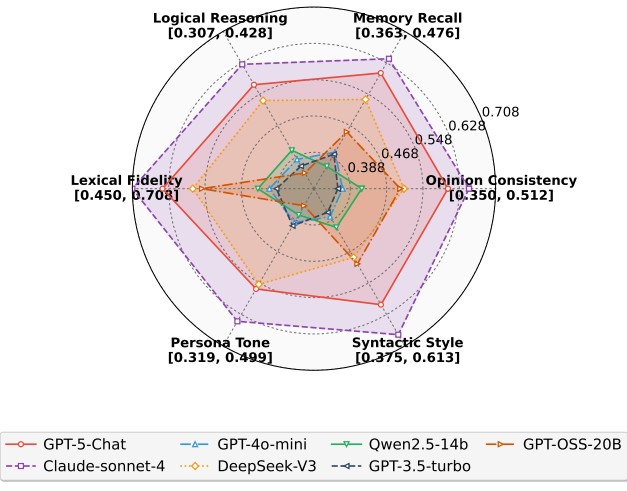

Figure 5: Dimension 1 (Social Persona): Discriminative evaluation radar (accuracy-based) across six capabilities (all labeled capabilities). Shows multiple-choice persona matching performance; higher is better.

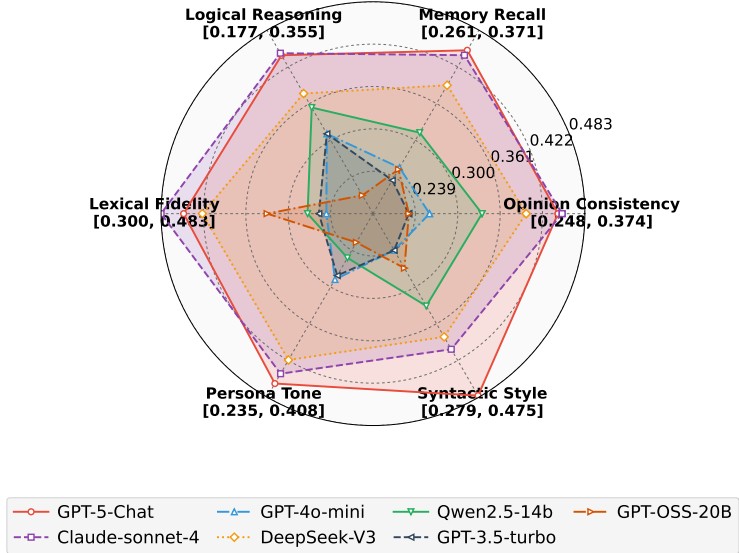

Figure 6: Dimension 1 (Social Persona): Generative Ranking radar (LLM-as-a-Judge, Acc.(Gen)) across six capabilities (all labeled capabilities). Reflects relative imitation quality; higher is better.

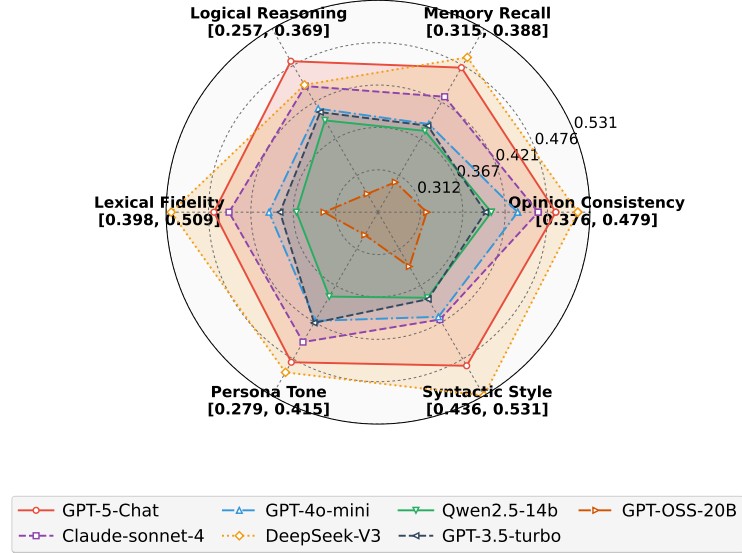

Figure 7: Dimension 1 (Social Persona): Generative Scoring radar (LLM-as-a-Judge, Score(Gen), 1–5) across six capabilities (all labeled capabilities). Captures absolute similarity to the ground truth; higher is better.

## D.2    INTERPERSONAL PERSONA (DIMENSION 2)

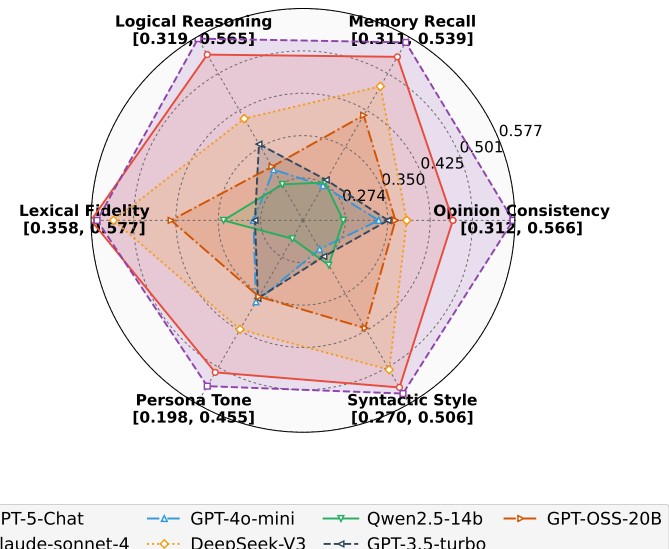

Figure 8: Dimension 2 (Interpersonal Persona): Combined Average radar over six capabilities (all labeled capabilities). Aggregates across discriminative and generative protocols; higher is better along each spoke.

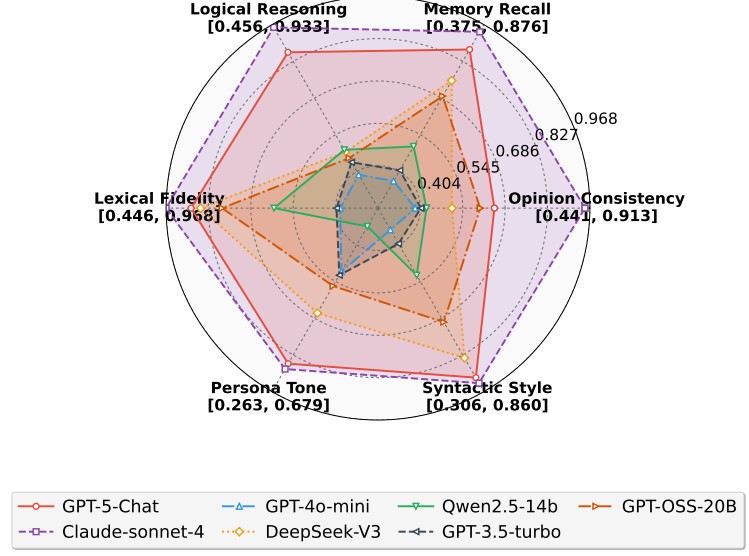

Figure 9: Dimension 2 (Interpersonal Persona): Discriminative evaluation radar (accuracy-based) across six capabilities (all labeled capabilities). Shows multiple-choice persona matching performance; higher is better.

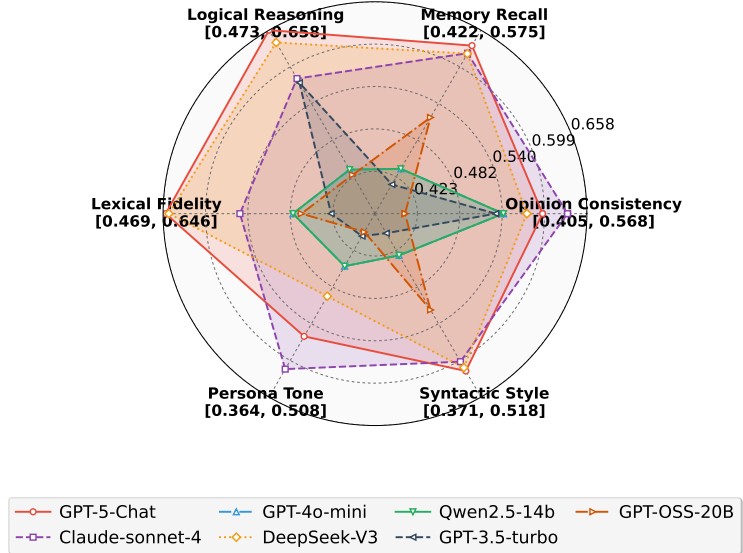

Figure 10: Dimension 2 (Interpersonal Persona): Generative Ranking radar (LLM-as-a-Judge, Acc.(Gen)) across six capabilities (all labeled capabilities). Reflects relative imitation quality; higher is better.

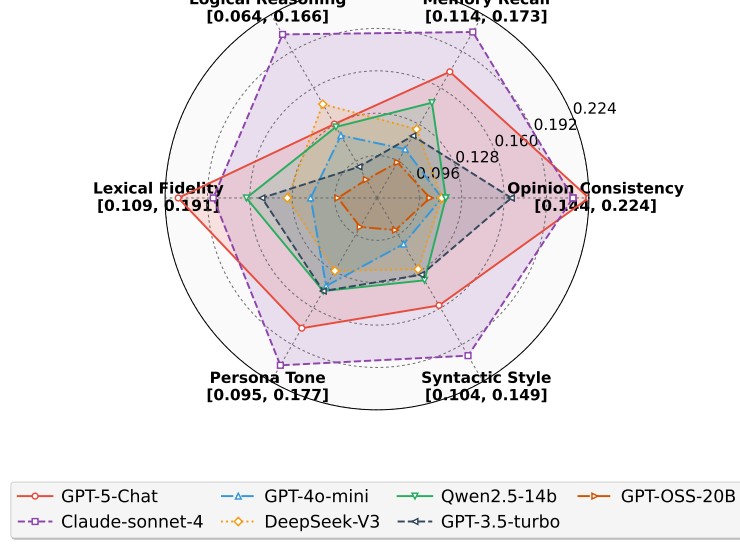

Figure 11: Dimension 2 (Interpersonal Persona): Generative Scoring radar (LLM-as-a-Judge, Score(Gen), 1–5) across six capabilities (all labeled capabilities). Captures absolute similarity to the ground truth; higher is better.

## D.3 NARRATIVE PERSONA (DIMENSION 3)

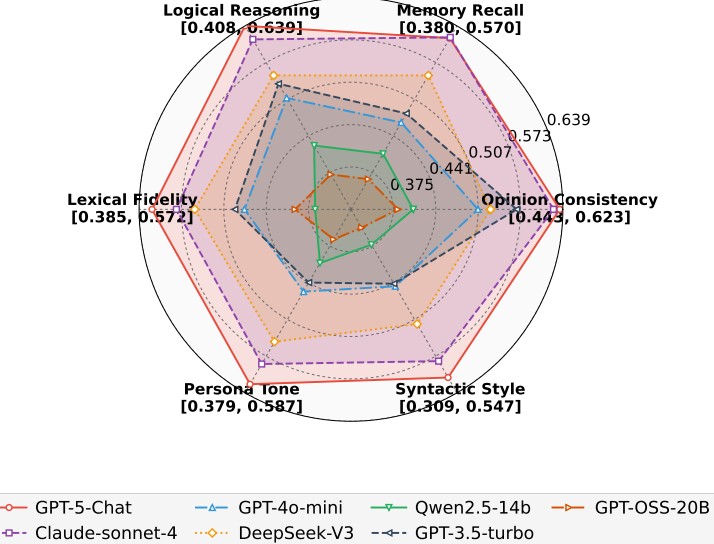

Figure 12: Dimension 3 (Narrative Persona): Combined Average radar over six capabilities (all labeled capabilities). Aggregates across discriminative and generative protocols; higher is better along each spoke.

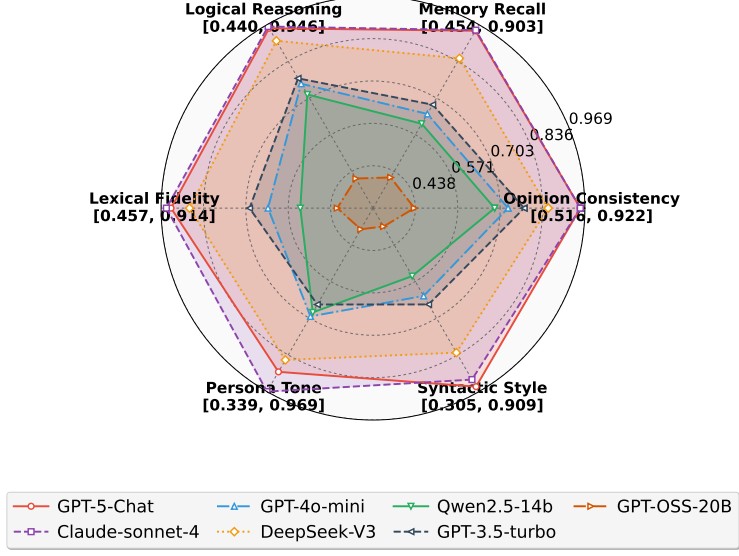

Figure 13: Dimension 3 (Narrative Persona): Discriminative evaluation radar (accuracy-based) across six capabilities (all labeled capabilities). Shows multiple-choice persona matching performance; higher is better.

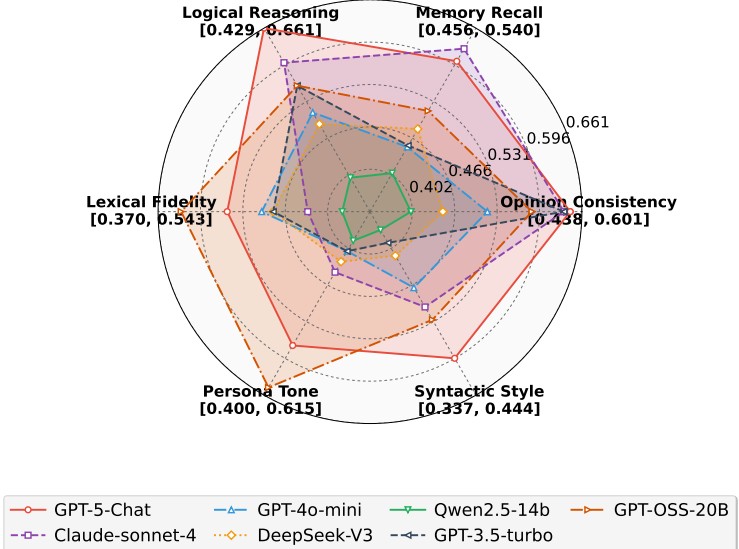

Figure 14: Dimension 3 (Narrative Persona): Generative Ranking radar (LLM-as-a-Judge, Acc.(Gen)) across six capabilities (all labeled capabilities). Reflects relative imitation quality; higher is better.

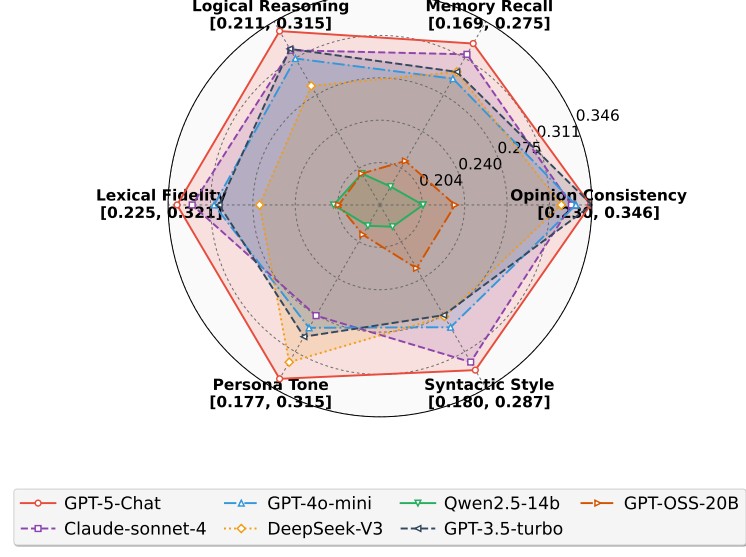

Figure 15: Dimension 3 (Narrative Persona): Generative Scoring radar (LLM-as-a-Judge, Score(Gen), 1–5) across six capabilities (all labeled capabilities). Captures absolute similarity to the ground truth; higher is better.

# E HUMAN ANNOTATION GUIDELINES

## E.1 TASK BACKGROUND AND OBJECTIVES

This study aims to evaluate the performance of Large Language Models (LLMs) as judges in digital twin tasks. To validate the reliability of model judgments, we need human annotators to independently annotate selected data to establish a trustworthy benchmark.

The annotation task consists of three subtasks corresponding to different evaluation modes: discriminative tasks, generative ranking tasks, and generative scoring tasks. Each annotator will annotate the same 100 data samples to ensure consistency and comparability in evaluation.

**Important Note**: All provided content (anchor posts, reply history, choices) is in Chinese. You should analyze and understand the content within the Chinese language context, but your reasoning and annotations should be provided in English when specified.

## E.2 DISCRIMINATIVE TASK ANNOTATION

### E.2.1 TASK DESCRIPTION

In the discriminative task, you need to act as a specific social media user, becoming their digital twin. Based on the given conversation history and anchor post, select the most appropriate reply from four candidates that best matches the user's personal style and language habits.

### E.2.2 LLM PROMPT (USE THE SAME EVALUATION STANDARD)

The LLM uses the following prompt for this task. Please follow the same reasoning approach:

> *Your task is to act as a specific social media user, becoming their digital twin. Note: All provided text (history, post, choices) is in Chinese. You must analyze the user's style directly within the Chinese language context.*
> Based on the user's reply history, think and respond with their mindset, tone, and style.
> Your reply history: (Note: "AnchorPost" is another user's post, and "UserReply" is your own reply.)
> Now, you see a new post: [anchor post]
> Below are 4 candidate replies. Which one is most likely something you would say?
> Please respond by explaining your choice from the user's perspective using "I".

### E.2.3 EVALUATION CRITERIA

- **Style Consistency**: Does the reply maintain consistency with the user's language style demonstrated in conversation history?
- **Tone Matching**: Does the reply's tone (formal/informal, humorous/serious, etc.) match the user's characteristics?
- **Vocabulary Usage**: Are the vocabulary choices and expressions consistent with the user's habits?
- **Logical Coherence**: Is the reply content logically related to the anchor post and historical context?

### E.2.4 ADDITIONAL HUMAN GUIDANCE

- Carefully read through the entire conversation history to understand the user's communication patterns
- Pay attention to recurring phrases, greeting patterns, and emotional expressions
- Consider the user's typical response length and level of detail
- Think from the user's perspective: "If I were this user, which response would I most likely choose?"

### E.2.5  ANNOTATION METHOD

Please fill in the option number (0, 1, 2, or 3) that you consider most appropriate in the `human_choice` field, corresponding to the index position in the choices array.

## E.3  GENERATIVE RANKING TASK ANNOTATION

### E.3.1  TASK DESCRIPTION

In the generative ranking task, you need to identify which candidate reply is most similar to a reference reply in terms of style, tone, vocabulary, sentiment, and topic.

### E.3.2  LLM PROMPT (USE THE SAME EVALUATION STANDARD)

The LLM uses the following prompt for this task:

> You are an expert evaluator of writing style. Your task is to compare several candidate replies against a known "Reference Reply" written by a specific user.
>
> Your goal is to identify which candidate is the most similar to the reference in terms of style, tone, vocabulary, sentiment, and topic.
>
> Now, determine which single candidate is the closest match to the Reference Reply. The reasoning should be concise, limited to 2-3 sentences, focusing on the stylistic similarities.

### E.3.3  EVALUATION CRITERIA

- **Style Similarity**: Lexical choices, sentence structure, formality level
- **Tone Matching**: Emotional tone, attitude, and mood
- **Vocabulary Consistency**: Use of similar words, phrases, or expressions
- **Sentiment Alignment**: Overall emotional orientation and sentiment
- **Topic Relevance**: Relevance and approach to the main topic

### E.3.4  ADDITIONAL HUMAN GUIDANCE

- Focus on stylistic elements rather than factual content
- Look for subtle language patterns and preferences
- Consider both what is said and how it is said
- Compare the "voice" and "personality" reflected in each candidate

### E.3.5  ANNOTATION METHOD

Please fill in the letter (A, B, C, or D) of the option you consider best matching in the `human_choice` field.

## E.4  GENERATIVE SCORING TASK ANNOTATION

### E.4.1  TASK DESCRIPTION

In the generative scoring task, you need to assess how well a generated reply replicates a ground truth reply, providing a score from 1-5 based on comprehensive evaluation criteria.

### E.4.2  LLM PROMPT (USE THE SAME EVALUATION STANDARD)

The LLM uses the following detailed evaluation framework:

> *You are a meticulous and objective evaluator for a digital twin benchmark. Your task is to assess how well a 'Generated Reply' replicates a 'Ground Truth Reply' for a given social media post.*

The evaluation rests on three key pillars:

1. **Opinion Consistency**: Does the Generated Reply express the exact same core opinion, stance, and sentiment as the Ground Truth?
2. **Logical & Factual Fidelity**: Is the Generated Reply based on the same reasoning and facts as the Ground Truth?
3. **Stylistic Similarity**: How closely does the Generated Reply match the Ground Truth in terms of lexical, tone, and syntactic elements?

### E.4.3 SCORING RUBRIC (1-5 SCALE)

- **5 - Perfect Replication**: Perfect match across all three pillars. Feels like a natural, alternative expression from the same user.
- **4 - High Fidelity**: Opinion and Logic/Factual pillars are perfectly matched. Only minor, subtle differences in Style.
- **3 - Core Alignment, Detail Loss**: Core opinion is consistent, but noticeable loss of detail in Logic or Style pillars.
- **2 - Partial Relevance, Major Deviation**: Major failure in at least one of the three pillars.
- **1 - Irrelevant or Contradictory**: Almost nothing in common with the Ground Truth or expresses contradictory opinion.

### E.4.4 ADDITIONAL HUMAN GUIDANCE

- First identify the core opinion/stance in the ground truth reply
- Check if the generated reply maintains the same logical flow and reasoning
- Evaluate stylistic elements: word choice, sentence length, formality, emotional tone
- Consider the reply as a whole - would it serve as an acceptable substitute?
- Be objective and consistent across all annotations

### E.4.5 ANNOTATION METHOD

Please fill in your score (1, 2, 3, 4, or 5) in the `human_score` field.

## E.5 GENERAL GUIDELINES AND NOTES

### E.5.1 QUALITY ASSURANCE

- Read all conversation history carefully to understand the user's communication patterns
- Maintain objectivity and consistency throughout the annotation process
- Avoid letting personal preferences influence your judgment
- Each data sample should be annotated independently
- When facing difficult decisions, choose the relatively best option
- Double-check for missing annotations or format errors after completion

### E.5.2 LANGUAGE CONSIDERATIONS

- All content is in Chinese - analyze within the Chinese language context
- Pay attention to Chinese-specific expressions, internet slang, and cultural references
- Consider Chinese punctuation and writing conventions
- Understand the social media context and communication norms

# F  USE OF LARGE LANGUAGE MODELS

## F.1  SCOPE OF USE

LLMs assisted with (i) prompt drafting and refinement, (ii) minor code refactoring suggestions, (iii) generating synthetic evaluation items (e.g., distractor options and candidate responses), and (iv) light copy-editing of non-technical prose. LLMs did *not* originate novel claims, conduct final analyses, or decide conclusions; all substantive results are author-verified.

## F.2  MODELS AND ACCESS

We used the following LLMs via API/local inference: **GPT-5-Chat** (OpenAI), **Claude-Sonnet-4** (Anthropic), **DeepSeek-V3** (DeepSeek), **GPT-4o-mini** (OpenAI), **GPT-3.5-Turbo** (OpenAI), **GPT-OSS-20B** (Open-source community), **Qwen2.5-14B** (Alibaba / Qwen Team). **Access window:** 06/2025–09/2025.

## F.3  HUMAN OVERSIGHT

All LLM outputs were screened by the authors; items entering quantitative evaluation were validated via deterministic scripts or double review.

## F.4  REPRODUCIBILITY

We include the full evaluation prompts and protocols, the 1–5 scoring rubric, the textual recipes for constructing multiple-choice questions, the data filtering thresholds per dimension, dataset sizes/statistics, and the evaluation equations and metrics. These disclosures are sufficient to re-implement our evaluation.

## F.5  DATA PRIVACY AND SAFETY

Only public data were processed; no PII or sensitive user data were sent to third-party services. We complied with provider Terms of Service and applied toxicity/safety filters where applicable.

## F.6  LIMITATIONS

LLM outputs may reflect training-data biases or hallucinations. We mitigated these via rule-based validators and manual review; residual errors may remain.

