# OpenReview forum: "TwinVoice: A Multi-dimensional Benchmark Towards Digital Twins via LLM Persona Simulation"
_ICLR.cc/2026/Conference — ICLR 2026 Conference Withdrawn Submission_

### Official Review · Reviewer_8Myh · 2025-10-29

**Soundness:** 1
**Presentation:** 2
**Contribution:** 2
**Rating:** 2
**Confidence:** 3

**Summary:**

This paper introduces TwinVoice, a benchmark for evaluating persona simulation using LLMs. TwinVoice includes three dimensions (settings or contexts for persona simulation) and six capabilities (evaluation criteria). TwinVoice sources its ground truth from real-world human writing across multiple languages and evaluates in both discriminative and generative answer formats. The empirical evaluations on various LLMs, including state-of-the-art LLMs, reveal that six capabilities for each LLM are correlated, while strengths and weaknesses across LLMs are stable.

**Strengths:**

1. This paper addresses an important problem of evaluating LLMs at human simulation.

2. The scaled-up evaluation protocol of TwinVoice provides more comprehensive information on the human-simulation performance of LLMs.

**Weaknesses:**

1. The paper does not provide evidence of the value this new benchmark adds over the existing benchmarks. It is unclear whether the findings in this paper could have been obtained using existing benchmarks and, if not, which component of this benchmark enabled it.

2. There is no clear rationale behind the choices of six capabilities. They seem to overlap, and not all of them will be relevant at every turn. For example, persona tone will correlate with both lexical and syntactic choices. And not all conversations involve logical reasoning.

3. Discriminative evaluation is not well justified in its practicality and value. The real-world simulation will mostly take a generative form, so it would be helpful to see if discriminative evaluation agrees with generative evaluation instance-wise. Moreover, there are insufficient details about the choice of distractors for discriminative evaluation, which is essential for contextualizing it.

**Questions:**

1. What are the findings that are not available in other benchmarks but TwinVoice enables? Would it require non-trivial effort to make similar findings on other benchmarks? Do you have supporting empirical evidence? What component of TwinVoice enables it?

2. How did the authors decide on the six capabilities? Are all instances of the data appropriate for evaluating these six capabilities? If not, do you have an estimate on how much of the data is relevant to each capability? For each capability, are there any deterministic or cheaper alternative metrics that can replace LLM-as-judge?

3. Is discriminative evaluation really needed in addition to generative evaluation? How were distractors selected?

---

### Official Review · Reviewer_3bXF · 2025-10-30

**Soundness:** 3
**Presentation:** 3
**Contribution:** 3
**Rating:** 6
**Confidence:** 4

**Summary:**

* This paper introduces the TwinVoice benchmark, a novel and large-scale benchmark for realistic and fine-grained LLM persona simulation.
* The work proposes a novel evaluation framework for persona fidelity, targeting analysis from 2 main aspects: mindset coherence and linguistic expression.
* The paper conducts comprehensive evaluation experiments on persona simulation performance with multiple state-of-the-art LLMs, revealing current limitations in aspects such as syntactic style and memory recall.

**Strengths:**

* The proposed TwinVoice benchmark is more comprehensive than previous works:
  * TwinVoice has over 4,500 personas for evaluation, exceeding the size of prior works. The persona dataset is dissected into 3 categories: social persona, interpersonal persona, and narrative persona. This improves the diversity and robustness of persona fidelity evaluation.
  * TwinVoice consists of both real-world and synthetic data, resolving the issue with dominating synthetic data usage in existing works.
* The proposed TwinVoice benchmark approaches persona simulation assessment from multiple aspects, offering new perspectives of evaluating personalized LLM systems.
  * The work proposes a novel evaluation framework for persona fidelity, targeting analysis from 2 main aspects: mindset coherence and linguistic expression.
    * The 2 aspects of persona simulation evaluation is further elaborated into the test of 6 fundamental capabilities, such as logical reasoning and opinion consistency for mindset coherence and lexical fidelity for linguistic expression.
  * Evaluation methodology includes both discriminative multiple-choice assessment and open-ended LLM-as-a-Judge paradigm.
    * The GPT5-as-a-judge evaluation framework is human-verified with decent agreement score.
* The paper conducts comprehensive evaluation experiments on persona simulation performance with multiple state-of-the-art LLMs, revealing current limitations in aspects such as syntactic style and memory recall.

**Weaknesses:**

* While I appreciate the authors for conducting a human study on the proposed LLM-as-a-judge evaluation framework, I still have minor concerns about the pipeline's robustness:
  * First, for the human verification scale, 50 items per judging mode (100 total) for such a big evaluation benchmark might not be enough.
  * Additionally, previous works [1] [2] have revealed robustness issues with LLM-as-a-judge frameworks, and similar biases could lead to robustness issues of the proposed evaluation framework.
    * For example, [1] discusses how LLM judges prefer texts that are more familiar to them (self-preference bias), like their own generations or generations from LLMs of similar architecture / training data. **This might explain the good performance of GPT-5-Chat (strongest aggregate generative performance as stated in line 374), since you are using GPT5 as the judge.**
    * Another example would be position bias [2]. **Permutating the choices in the multiple choice & ranking evaluation framework and take the aggregated result** will bring more robustness.
* Lack of qualitative analysis on experiment results, especially on aspects where models fail (e.g. syntatctic style, memory recall). Looking at the numbers and charts, I have a hard time understanding what failure pattens we can observe in models' generation outputs.

[1] Wataoka, Koki, Tsubasa Takahashi, and Ryokan Ri. "Self-Preference Bias in LLM-as-a-Judge." Neurips Safe Generative AI Workshop 2024.
[2] Thakur, Aman Singh, et al. "Judging the judges: Evaluating alignment and vulnerabilities in llms-as-judges." arXiv preprint arXiv:2406.12624 (2024).

**Questions:**

* How robust is the proposed LLM-as-a-Judge pipeline against issues like position and self-preference biases?
* Can you show some failure cases that show model failure modes in a more straightforward way?

---

### Official Review · Reviewer_z3pb · 2025-10-31

**Soundness:** 2
**Presentation:** 1
**Contribution:** 2
**Rating:** 2
**Confidence:** 5

**Summary:**

Please see the weakness section.

**Strengths:**

Please see the weakness section.

**Weaknesses:**

**Significant ICLR formatting violation**

This submission appears to use 1.0 inch left / right margins, significantly below the regulation of 1.5 inch (“Formatting instructions for ICLR 2026 conference submissions,” Line 30, Line 50). This expands the text width from the mandated 5.5 inch to 5.5 + 0.5 * 2 = 6.5 inch, so 9 pages × (6.5 / 5.5) = 10.64 pages of effective content, exceeding the strict 9-page limit (“At the time of submission, the main text should be 9 pages or fewer… This limit will be strictly enforced. Papers with main text beyond the page limit will be desk-rejected.” of the ICLR 2026 author guide https://iclr.cc/Conferences/2026/AuthorGuide). The “Formatting instructions for ICLR 2026 conference submissions” also warns that “tweaking the style files may be grounds for rejection.”

Historically, there has been papers desk-rejected for the exactly same problem.

- “Circuit Compositions: Exploring Modular Structures in Transformer-Based Language Models”, ICLR 2025 submission https://openreview.net/forum?id=u4XyECA6Zd)

- “EVLM: An Efficient Vision-Language Model for Visual Understanding”, ICLR 2025 submission https://openreview.net/forum?id=S7M1iqFLVm

- “Inductive Bias of Multi-Channel Linear Convolutional Networks with Bounded Weight Norm”, ICLR 2022 submission https://openreview.net/forum?id=NMSugaVzIT

**Questions:**

Please see the weakness section.

---

> ### Author Response · Authors · 2025-11-14
> **Response to Reviewer z3pb's Formatting Concern**
>
> Dear Reviewer z3pb,
>
> Thank you for identifying the formatting issue in our submission. We acknowledge that our original submission used incorrect margins (1.0 inch instead of 1.5 inch) and have corrected this issue.
>
> **Corrective action:**
>
> We have uploaded a revised version (November 14, 2025) with:
> - Correct 1.5-inch margins using unmodified official ICLR 2026 style files
> - 10 pages of main text, complying with the Author Guide: "During the discussion/rebuttal phase and for the camera-ready, the page limit will be increased to 10 pages to allow for new results/discussions."
>
> According to ICLR guidelines: "If reviewers identify violations (e.g., plagiarism, double submission, paper length, formatting, etc.), they should contact either the AC/SAC or the PC as appropriate."
>
> We would be grateful if you could engage with the scholarly aspects of our work. We are carefully addressing the technical feedback from other reviewers and believe our research makes meaningful contributions.
>
> Thank you for your attention to this matter.
>
> Sincerely,
> Authors

---

> > ### Comment · Reviewer_z3pb · 2025-11-14
> >
> > I cannot proceed with the review, as the paper already violated the author guidelines in a way that constitutes grounds for desk rejection.
> >
> > The fact that the revised version now fits within the page limit does not, in my view, justify reversing that decision. It would be unfair to accept this paper while others have been desk rejected for the exactly same issues (and to my knowledge, there has been no policy change regarding incorrect page margins from past years).
> >
> > Dear authors, I could not find guideline stating that “If reviewers identify violations (e.g., plagiarism, double submission, paper length, formatting, etc.), they should contact either the AC/SAC or the PC.” Could you point me to the reference?

---

> > > ### Author Response · Authors · 2025-11-15
> > > **Response to Reviewer z3pb**
> > >
> > > Dear Reviewer z3pb,
> > >
> > > Thank you for your response.
> > >
> > > Regarding the guideline reference you requested, please refer to the **ICLR 2026 Reviewer Guide** at https://iclr.cc/Conferences/2026/ReviewerGuide
> > >
> > > Specifically, in the **FAQ for Reviewers**, Question 5 states:
> > >
> > > **Q: How should I handle a policy violation?**
> > >
> > > **A:** To flag a CoE violation related to a submission, please indicate it when submitting the CoE report for that paper. The AC will work with the PC and the ethics board to resolve the case. **To discuss other violations (e.g. plagiarism, double submission, paper length, formatting, etc.), please contact either the AC/SAC or the PC as appropriate.** You can do this by sending a confidential comment with the appropriate readership restrictions.
> > >
> > > We acknowledge the formatting issue in our original submission and have corrected it. We understand your concern about fairness. However, we respectfully note that:
> > >
> > > 1. Our submission was not desk-rejected during the initial screening phase
> > > 2. We have now uploaded a compliant version
> > > 3. The appropriate venue for determining whether this constitutes grounds for desk rejection is through the AC/PC process, as outlined in the reviewer guidelines
> > >
> > > We appreciate your diligence and completely understand that the final decision rests with the Area Chair and Program Chairs according to ICLR procedures.
> > >
> > > We would be grateful if you could proceed with reviewing the scholarly content of our work while the AC addresses the procedural matter.
> > >
> > > Sincerely,
> > > Authors

---

### Official Review · Reviewer_CJUf · 2025-11-01

**Soundness:** 2
**Presentation:** 2
**Contribution:** 2
**Rating:** 4
**Confidence:** 3

**Summary:**

This paper suggests a new benchmark that assesses LLM's ability of replicating human persona. The authors used three different datasets to model human persona in terms of three aspects: social, interpersonal, and narrative. To ensure the validity and generalizability of experiment, the authors measured three aspects with three different methods: discriminative, generative(scoring) and generative(ranking). Also, they further measured each aspect in terms of six capabilities. The result found that language models have some ability to replicate human persona, but there's still room for improvement, especially on capabilities of memory recall and persona-tone alignment. And, the authors also noted that the gap between discriminative and generative evaluation protocols indicates the difficulty of open-ended persona replication.

**Strengths:**

- The paper could provide a framework for detailed analysis on digital twin (or human persona replication).
- The paper provides some discussion about the strength and weakness of present LLMs.
- The paper tested multiple models, which strengthens its generalizability.

**Weaknesses:**

- The experimental method has issues. The selection of language models is not systematic (and there's no reason specified in the paper), and the judge model overlaps with the generation model. The agreement measurement is not proper. See Question A.
- The paper's review on the previous benchmark or papers is somewhat shallow. Other researchers have been reported similar findings, especially on memory recall and persona consistency. Though the paper aims to be a unified framework, the paper should discuss the difference between it and the other previous work clearly to avoid confusion of reader. See Question B.
- The paper's discussion is not deep enough. To provide some insight for the community, this paper could provide some reason or conjecture which can guide the future research. The current depth is insufficient regarding this aspect. See Questions A and B.

**Questions:**

## Question A. Experimental method

A1. Why did the authors selected those seven models? What is the main criteria of selection?
A1-1. If there exists some criterion, could the authors link the findings with those criteria? Are there any factors affected the result?

A2. The authors used GPT-5 for LLM-as-a-judge and GPT-5-Chat for answer generation. As the model stems from the same training data distribution, the judge might prefer its family which can show a similar generation behavior to them. Why did the authors selected same model for both judge and generation?
A2-1. Does this selection affect the result? How?

A3. The authors used Gutenberg project, which is a very well known corpus in the community. As the corpus was commonly used for training language models, it is highly likely that the models already learned the corpus. Doesn't this possibly affect the result for narrative aspect?
A3-1. Can we check whether the model already learned the corpus, using the methods for checking data contamination?

A4. This is a minor but could highly affect the result on human evaluation. The authors mentioned that they used three annotators. But they used Cohen's kappa, which is limited to two annotators because of its statistical assumption. In the case of three or more annotators, researchers usually adopt Fleiss' kappa, which is an extension of Cohen's kappa. So the question is that how did the authors provide the result of kappa? Were the authors averaged the result of all pairs?
A4-1. For more statistically sound result, could the authors provide Fleiss' kappa instead of Cohen's?

A5. Though the authors used temperature 0, it does not ensure deterministic behavior. Did the authors run the experiment multiple times? If so, could the authors provide the statistical errors or other statistical comparison results to support their discussion?
A5-1. It could be better provide the result of statistical tests.

A6. Lastly, how the authors provide history to generate chat in interpersonal experiment? Did the authors input all elements in the history?
A6-1. Studies in persona-based dialogue generation usually use RAG or summarization systems to compact the history. If the authors used the entire history without summarization, can the difference affect the result? How?

## Question B. Previous work

B1. This question is in the same line with Question A6. Previous methods on persona-based dialogue generation use different methods. As these methods might affect the result, the authors should shortly and clearly state the difference between their method and the recent advances. For example, please refer to [1].

[1] Saber Zerhoudi et al., PersonaRAG: Enhancing Retrieval-Augmented Generation Systems with User-Centric Agents, https://arxiv.org/abs/2407.09394

B2. Previously, researchers have discussed specific capabilities. What is the difference between this work and theirs, except for building a summative framework? For example, please refer to [1], [2], [3], and [4]. Note that these are few of related studies; there are lot of relevant studies.
B2-1. Could the findings be connected with these studies?

[2] Junhyuk Choi et al., Examining Identity Drift in Conversations of LLM Agents, https://arxiv.org/abs/2412.00804
[3] J. Huang et al., On the Humanity of Conversational AI: Evaluating the Psychological Portrayal of LLMs, ICLR 2024
[4] R. Chen et al., Learning to Memorize Entailment and Discourse Relations for Persona-Consistent Dialogues, AAAI 2023

## Question C. Other

C1. In terms of presentation, Figure 3 is not appropriate if the authors want to compare between different capabilities of the same model. Redrawing the figure by grouping them with model, instead of capabilities might be more suitable; especially for lines 363-364.

C2. I think the authors should warn the readers that their framework might be used for replicating a human behavior and can be used for fraud. This statement should be added in the ethics statement.

---

### Note · Authors · 2025-11-24

I have read and agree with the venue's withdrawal policy on behalf of myself and my co-authors.